# Molecular characterization of *Chlamydomonas reinhardtii* telomeres and telomerase mutants

Stephan Eberhard[1], Sona Valuchova[3], Julie Ravat[1], Jaroslav Fulneček[3], Pascale Jolivet[2], Sandrine Bujaldon[1], Stéphane D Lemaire[2], Francis-André Wollman[1], Maria Teresa Teixeira[2], Karel Riha[3], Zhou Xu[2,4]

Telomeres are repeated sequences found at the end of the linear chromosomes of most eukaryotes and are required for chromosome integrity. Expression of the reverse-transcriptase telomerase allows for extension of telomeric repeats to counteract natural telomere shortening. Although *Chlamydomonas reinhardtii*, a photosynthetic unicellular green alga, is widely used as a model organism in photosynthesis and flagella research, and for biotechnological applications, the biology of its telomeres has not been investigated in depth. Here, we show that the *C. reinhardtii* (TTTTAGGG)$_n$ telomeric repeats are mostly nondegenerate and that the telomeres form a protective structure, with a subset ending with a 3′ overhang and another subset presenting a blunt end. Although telomere size and length distributions are stable under various standard growth conditions, they vary substantially between 12 genetically close reference strains. Finally, we identify *CrTERT*, the gene encoding the catalytic subunit of telomerase and show that telomeres shorten progressively in mutants of this gene. Telomerase mutants eventually enter replicative senescence, demonstrating that telomerase is required for long-term maintenance of telomeres in *C. reinhardtii*.

## Introduction

Photosynthetic algae are in the highlight of basic and applied research, not only because of their core role for Earth's biosphere in oxygen evolution and carbon fixation but also because of their increased use in biotechnology for the production of proteins, bulk chemicals, and high-value molecules (Scaife et al, 2015; Scranton et al, 2015). Thus, a detailed understanding of algal physiology, including their cell cycle, cell growth, and genome integrity, is of critical importance. *Chlamydomonas reinhardtii*, also referred to as the "photosynthetic yeast" (Rochaix, 1995), is the most prominent model organism in the green algae lineage. It is widely used for

biotechnological applications as well as to study fundamental processes, such as photosynthesis and cilia structure and function (Harris, 2001; Sasso et al, 2018). It has a fully sequenced nuclear genome of 111 Mb distributed over 17 chromosomes (Merchant et al, 2007) and is amenable to powerful genetic approaches (Harris, 2009). Although genetic transformation is available in this organism, targeted nuclear genome modification is still not straightforward, but the emerging CRISPR/Cas9 technology might help solve this limitation (Findinier et al, 2019).

In eukaryotes, telomeres are repeated sequences found at the extremities of linear chromosomes. They are important for chromosome integrity and may limit cell proliferation capacity in some organisms. By progressively shortening with each cell cycle because of the end replication problem, telomeres eventually become too short and trigger a cell cycle arrest termed replicative senescence (Lundblad & Szostak, 1989; Harley et al, 1990). Most unicellular eukaryotes and germ, stem, and cancer cells in multicellular organisms counteract telomere shortening by expressing telomerase, an enzyme that adds de novo telomere sequences and allows for an unlimited proliferation potential (Pfeiffer & Lingner, 2013; Wu et al, 2017). Despite the crucial functions of telomeres and telomerase in maintaining genome stability and controlling cell proliferation in many model organisms, including plants, ciliates, fungi, and mammals (Fulcher et al, 2014), telomere biology in algae remains to be investigated in depth.

To our knowledge, only a handful of studies on *C. reinhardtii* telomeres have been published. Early studies published in the 90s showed that (i) *C. reinhardtii* telomeres are composed of TTTTAGGG repeats, which are different from the *Arabidopsis*-type TTTAGGG sequence (Petracek et al, 1990); (ii) the size of cloned telomeric repeats ranges from 300 to 600 bp (Petracek et al, 1990; Hails et al, 1995); (iii) they form G-quadruplex structures in vitro (Petracek & Berman, 1992); and (iv) the Gbp1 protein binds in vitro to single-stranded telomere sequences through two RNA recognition motifs, with a preference for RNA when Gbp1 is monomeric and for DNA when it is dimeric (Petracek et al, 1994; Johnston et al, 1999). More

[1]Sorbonne Université, CNRS, UMR 7141, Institut de Biologie Physico-Chimique, Biologie du Chloroplaste et Perception de la Lumière chez les Micro-algues, Paris, France [2]Sorbonne Université, PSL Research University, CNRS, UMR 8226, Institut de Biologie Physico-Chimique, Laboratoire de Biologie Moléculaire et Cellulaire des Eucaryotes, Paris, France [3]Central European Institute of Technology, Masaryk University, Brno, Czech Republic [4]Sorbonne Université, CNRS, UMR 7238, Institut de Biologie Paris-Seine, Laboratory of Computational and Quantitative Biology, Paris, France

Correspondence: stephan.eberhard@ibpc.fr; zhou.xu@sorbonne-universite.fr

recently, bioinformatics studies focused on the evolutionary relationships of telomere sequences in green algae (Fulnečková et al, 2015, 2012). Finally, a broad study of telomerase activity in green algae revealed that telomerase activity in *C. reinhardtii* extracts is low or not detectable (Fulnečková et al, 2013).

To gain a better understanding of *C. reinhardtii* telomere structure and maintenance, we investigated telomere sequence and end structure, analyzed telomere length distribution across different reference strains, identified *CrTERT*, the gene encoding the catalytic subunit of telomerase, and provided a genetic analysis of telomerase function, thus opening new avenues of research on telomere dynamics, proliferation potential, and genome integrity in *C. reinhardtii*.

# Results

### *C. reinhardtii* telomeric repeats are mostly nondegenerate with few low-frequency variants

In their seminal article, Petracek et al (1990) cloned and sequenced a limited number of *C. reinhardtii* telomeric repeats, revealing their canonical TTTTAGGG sequence (Petracek et al, 1990). Telomeric repeats are also identifiable in 18 of 34 chromosome ends on the available v5.5 genome sequence of *C. reinhardtii* (https://phytozome.jgi.doe.gov; Fig S1A). As the sequenced genome shows some telomeric repeat variations, we analyzed telomeric repeat sequences on a larger scale and looked for putative variants of the canonical telomere sequence. We amplified telomeres by a PCR-based method (Forstemann et al, 2000) using a forward primer specific to a conserved subtelomere–telomere junction common to 10 telomeres from eight different chromosomes (Fig S1A and B). The reverse primer was universal and annealed to a sequence of cytosines artificially added at the 3′ end of the telomeres by terminal transferase reaction. After cloning into a plasmid and sequencing, we analyzed 32 telomere sequences, encompassing 709 repeats. We found that ~90% (n = 636) of the repeats corresponded to the canonical sequence TTTTAGGG. We also detected variants such as TTTAGGG (corresponding to the canonical *Arabidopsis thaliana* sequence, n = 37; either at the subtelomere–telomere junction, n = 24; or elsewhere, n = 13) or TTTTTAGGG (n = 13) and TTTTGGG (n = 8) (Table 1 and Fig S1B). These three variants were found in at least two independent clones at the same position in the telomere sequence, thus likely representing true low-frequency variants and not sequencing errors. We also detected sequence variants that occurred

only in single clones (n = 15) and for which PCR and/or sequencing errors can, therefore, not be ruled out. We conclude that *C. reinhardtii* telomeric repeats are mostly nondegenerate with few low-frequency variants.

### *C. reinhardtii* telomeres form a protective structure and a subset ends with a 3′ overhang, whereas another subset bears blunt ends

The protective structure formed by telomeric DNA bound by specific proteins is critical for telomere functions (Palm & de Lange, 2008). To test the presence of such a structure at *C. reinhardtii* telomeres, we performed a micrococcal nuclease (MNase) digestion of nuclei and asked whether telomere DNA would be protected from its activity. When nuclei were subjected to increasing amounts of MNase, nucleosomal DNA was protected from digestion and migrated at ~150 bp based on ethidium bromide staining (Fig 1A, left), as expected (Clark, 2010). Intermediate digestion products migrated in a typical ladder pattern corresponding to di-nucleosomes, tri-nucleosomes, and higher order structures (Fig 1A, left and middle, asterisks in the lane with one unit of MNase). Strikingly, Southern blot hybridization with a radioactive telomeric probe revealed a diffuse pattern, suggesting that telomeric DNA was protected from MNase digestion in a noncanonical manner (Fig 1A, right). As a control, the same membrane was stripped and probed for 18S rDNA, which showed the canonical nucleosome structure (Fig 1A, middle, asterisks). The size of the protected telomeric DNA was in the range of 200–700 bp, which could correspond to the full telomere length. This result suggests that telomeric DNA might be fully associated with and protected by a noncanonical nucleosomal structure or by other protein complexes, similar to telosomes as observed in yeasts, for example (Wright et al, 1992; Greenwood et al, 2018).

The chromosome end structure determines the protection strategies used to cap the telomere. In many species, telomeres end with a 5′ to 3′ single-stranded overhang, important for the t-loop structure in human telomeres, telomerase recruitment, and binding of specific capping proteins, such as the CST and Ku complexes and POT1 (Palm & de Lange, 2008; Giraud-Panis et al, 2010; Wellinger & Zakian, 2012). As it was reported that the Gbp1 protein preferentially binds single-stranded *C. reinhardtii* telomeric DNA (Johnston et al, 1999), the presence of a 3′ overhang would be consistent with a role of Gbp1 at telomeres, possibly protecting them from degradation and fusions similarly to telomere capping proteins in other species. To experimentally test the presence of a 3′ overhang at *C. reinhardtii* telomeres, we performed primer extension telomere repeat amplification (PETRA) (Heacock et al, 2004). PETRA requires the annealing of an adaptor primer (PETRA-T) to the overhang. After primer extension, the telomere was PCR-amplified using a unique subtelomeric forward primer and a reverse primer (PETRA-A) complementary to a tag sequence present in PETRA-T (Figs 1B and S1C). Successful amplification by PETRA is indicative of the presence of a 3′ overhang. Using primers specific for three different telomeres (1R, 9R, and 10R), we found robust amplification of PETRA products in three *C. reinhardtii* strains (CC4350+, T222+, and CC125+), strongly suggesting that these telomeres have a 3′ overhang of at least 12 nucleotides, corresponding to the size of the annealed part of PETRA-T to the overhang (Figs 1B and S1C). Control with prior

**Table 1.** Frequency of telomeric repeat motifs determined by telomere PCR and sequencing of 32 independent clones.

| Sequence | n | Frequency |
|----------|-----|-----------|
| TTTTAGGG | 636 | 89.7% |
| TTTAGGG | 37 | 5.2% |
| TTTTTAGGG | 13 | 1.8% |
| TTTTGGG | 8 | 1.1% |
| Others | 15 | 2.1% |

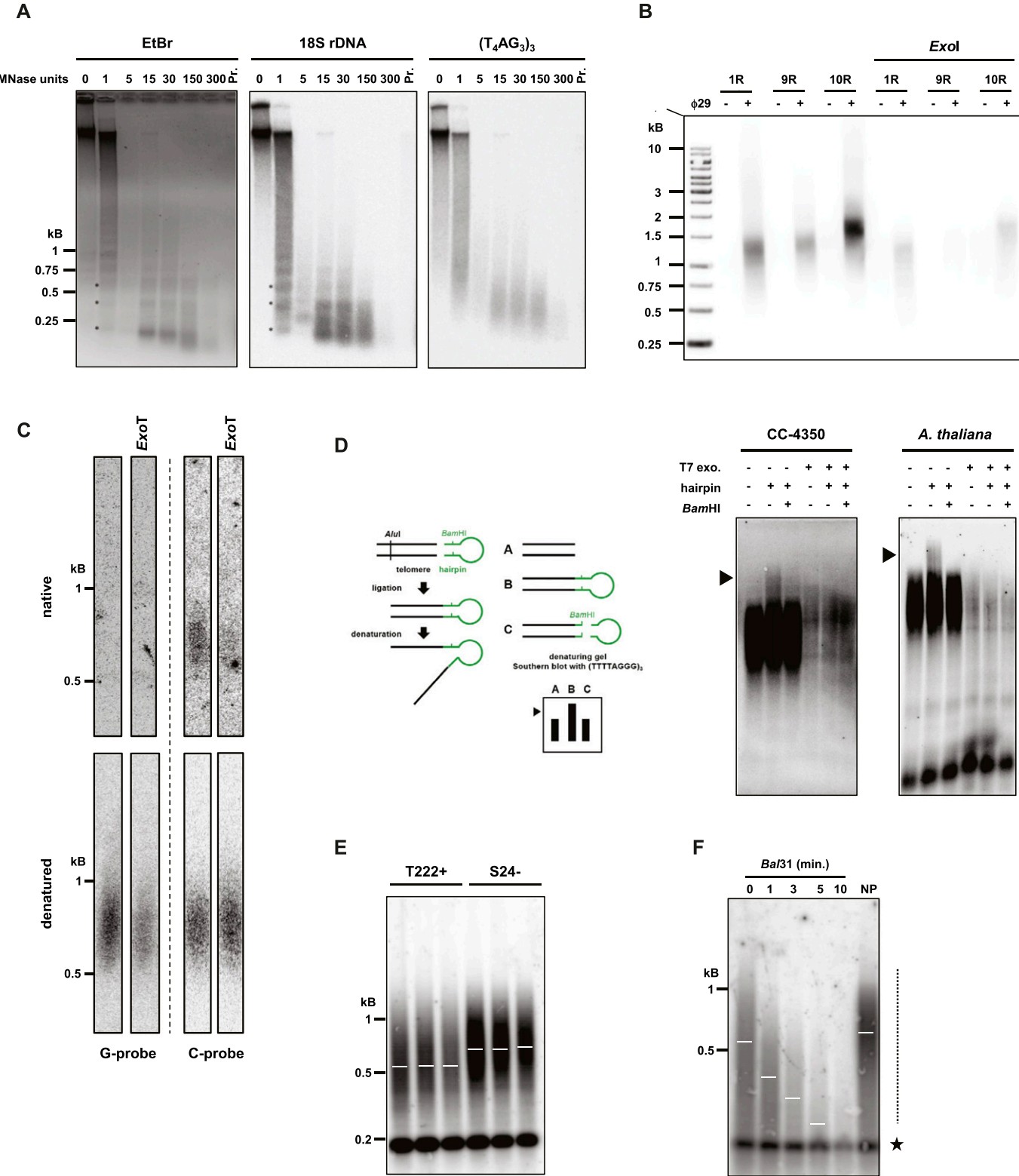

**Figure 1. Structural characterization of *C. reinhardtii* telomeres.**
**(A)** Characterization of *C. reinhardtii* telosomes by MNase digestion of chromatin (left panel; "EtBr": ethidium bromide staining of the migration gel) and Southern analysis with a telomeric specific probe (right panel; (TTTTAGGG)₃: radiolabeled probe). The membrane was then stripped and rehybridized with an 18S rDNA probe (middle panel). Lanes correspond to increasing amounts of MNase units; "Pr": control where nuclei were digested with 60 MNase units after protein removal. **(B)** PETRA was used to amplify three different telomeres (1R, 9R, and 10R) from strain CC4350+ and analyzed by Southern blot hybridization using the telomere-specific probe (TTTTAGGG)₃ (see also Fig S1C). Negative controls include omission of the primer extension step by Φ29 polymerase and pretreatment of the samples with *Exo*I. **(C)** Upper panel: Native

treatment of the samples with exonuclease I (*Exo*I) showed a decreased PETRA efficiency, indicating that the PETRA-T primer indeed requires 3′ overhang for annealing. The presence of a 3′ overhang at telomeres was further confirmed by native in-gel hybridization assay, which detected exonuclease T (*Exo*T)–dependent single-stranded DNA on the G-rich strand of the telomeres of strain T222+, but not on the C-rich strand (Figs 1C and S1D).

As it was shown that a subset of *A. thaliana* telomeres displays blunt ends instead of 3′ overhangs (Kazda et al, 2012), we asked whether blunt-ended telomeres also exist in *C. reinhardtii* because the PETRA and in-gel experiments do not exclude this possibility. To test this, we applied the hairpin assay, which was successfully used in *A. thaliana* to detect blunt-ended telomeres (Kazda et al, 2012). Briefly, a synthetic hairpin DNA can be ligated to both strands of the telomeres, only if they are blunt ended. After digestion with *Alu*I at a site in the subtelomeres, the ligated products migrate as a double-sized fragment compared with the unligated control in denaturing conditions. Cleavage of the ligated product by *Bam*HI, using a restriction site designed in the hairpin, can then show that the slow migrating product was indeed generated by ligation to the hairpin (Fig 1D, left). Ligation products of a higher molecular weight were clearly detected using this hairpin assay in *C. reinhardtii* strain CC4350+ as in *A. thaliana* (Fig 1D, right, arrows) and they were abolished by cleaving the hairpin by *Bam*HI or pretreatment with T7 exonuclease that generates 3′ single-stranded DNA protrusions at DNA ends. This demonstrates that a fraction of telomeres in *C. reinhardtii* is blunt ended.

Taken together, our structural analysis of telomeres indicates that as in *A. thaliana*, chromosome ends in *C. reinhardtii* are composed of two subsets, one ending with a 3′ overhang and the other with a blunt end.

## Terminal restriction fragment (TRF) analysis of *C. reinhardtii* telomeres

To study telomere length distributions and their possible regulations, we optimized a TRF analysis for *C. reinhardtii* to accurately measure telomere length from populations of cells.

We first measured telomere length in three independent biological replicates of strains T222+ and S24−, two isogenic reference strains differing only in their mating type (Gallaher et al, 2015). We found that telomere fragments spread as a smear over a large range of lengths, from ~200 to ~1,200 bp (Fig 1E). The two strains displayed a significant difference in their average telomere length (mean ± SD: T222+ = 539 ± 54 bp; N = 18 and S24− = 710 ± 12 bp; N = 5). To demonstrate that the detected smeary signal corresponded to terminal fragments, we digested the genomic DNA (gDNA) with exonuclease *Bal*31 before restriction digestion (Fajkus et al, 2005)

and indeed observed that with increasing incubation times with *Bal*31, the signal progressively decreased in size until it nearly disappeared after 10 min (Figs 1F and S1E). The migration of a band at ~200 bp was not altered even with the longest *Bal*31 treatment, indicating that it stemmed from interstitial telomere repeats located within the genome. Because this sharp band did not cross-react with a probe targeting TG microsatellite sequences (Fig S1F), it most probably corresponded to bona fide telomere sequence–containing region(s) of the genome and not to nonspecific cross hybridizations.

## Telomere length distribution is stable in different standard growth conditions

*C. reinhardtii* has been widely used as a model organism to study photosynthetic processes because of its ability to grow in different metabolic regimes (Harris, 2009). Under strictly phototrophic conditions (minimum medium in the light), photosynthesis is the only metabolic process providing ATP and reducing power to growing cells. In strictly heterotrophic conditions in the dark, *C. reinhardtii* can survive by respiring the acetate contained in Tris-acetate-phosphate (TAP) medium. In mixotrophic conditions, that is, TAP medium in the light, cells use a combination of photosynthesis and respiration to grow. Because in other organisms, environmental conditions can regulate telomere length (Walmsley & Petes, 1985; von Zglinicki, 2000; Epel et al, 2004; Romano et al, 2013), we asked whether telomeres vary in length and/or size distributions in response to different standard growth conditions.

We first tested whether cells displayed different telomere lengths during a standard growth kinetic in TAP medium, from inoculation to exponential and then stationary phase, sampled at different time points over a period of 8 d. We observed no significant difference in telomere length between the samples (Fig 2A). Prolonged incubation in stationary phase for up to 15 d also did not affect telomere length (Fig 2B). Thus, telomere length was not altered either in exponential growth in replete medium or in the absence of growth, during nutrient depletion, and with any other property of saturated cultures, even over a prolonged period.

We also asked whether stimulating cell growth could affect telomere length. Because of the multiple fission mode of cell division of *C. reinhardtii* (Cross & Umen, 2015), actively growing cells might spend less time in each phase of the cell cycle, and we reasoned that on average, telomerase might thus be less active. To test this hypothesis, a TAP culture was constantly maintained in exponential growth phase by serial dilutions over a period of 10 d. Telomere length did not significantly change (Fig 2C), and therefore, high division rate did not affect telomere length or distribution.

Finally, we checked telomere length distributions in cultures grown in either strictly phototrophic, strictly heterotrophic, or

---

in-gel hybridization assay of telomeres from strain T222+ using a G-probe (oT0958, left) or a C-probe (oT0959, right). Most of the native signal, when hybridized with the C-probe, was absent when the gDNA was pretreated with *Exo*T. Lower panel: The native gel was then denatured and transferred to a membrane, which was then hybridized with the same probes. The uncropped gel and membrane are shown in Fig S1D. **(D)** Hairpin ligation assay on *C. reinhardtii* CC4350+ strain and *A. thaliana*. A scheme of the assay is shown (left). Digestion with *Bam*HI, which removes the ligated hairpin and pretreatment with T7 exonuclease ("T7 exo."), which resects the 5′ end of a duplex DNA, are used as controls. **(E)** T222+ and S24− strains were subcloned and three subclones were independently grown in liquid culture until stationary phase and subsequently analyzed by TRF Southern blot hybridization. **(F)** gDNA was subjected to *Bal*31 digestion for 0 to 10 min. Digested products were column-purified and then processed for TRF analysis. 0: no *Bal*31 digestion, but gDNA was column-purified before digestion by the restriction enzymes. NP: gDNA was directly analyzed by TRF, with No column Purification. Dashed line: smear corresponding to telomeres. Star: *Bal*31-insensitive band, corresponding to interstitial telomeric repeat (see also Fig S1E).

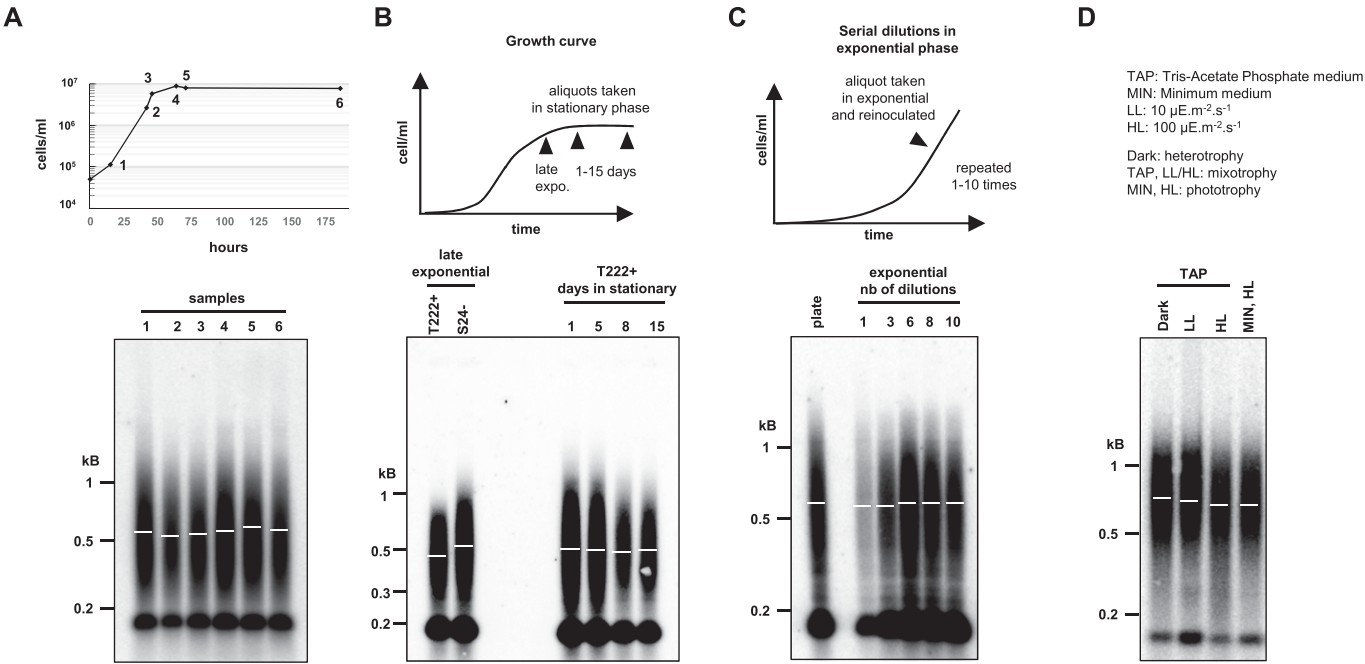

**Figure 2. Telomere length distribution is stable under various growth conditions.**
**(A)** Telomere length distributions of T222+ strain at different growth stages of liquid cultures. T222+ cells were harvested at early exponential (1), mid-exponential (2), late exponential (3), and early (4, 5) and late (6) stationary phases and analyzed by TRF Southern blot hybridization. **(B)** Telomere length distributions of prolonged cultures in stationary phase. The cells were harvested after 1, 5, 8, and 15 d after reaching stationary phase and compared with late exponential cultures. **(C)** Telomere length distributions of serial dilutions of rapidly growing cells. A liquid culture of T222+ cells was grown to exponential phase ($2 \times 10^6$ cells/ml), a sample of cells was harvested and the remaining cells diluted with fresh media to $5 \times 10^4$ cells/ml. This serial dilution was repeated 10 times. Samples corresponding to dilutions 1, 3, 6, 8, and 10 were then analyzed by TRF Southern blot hybridization. Plate: Cells were directly scraped from 1-wk-old streaks on TAP Petri dishes, without liquid culture. **(D)** Telomere length distributions in different metabolic growth conditions. The cells were grown for 6 d to stationary phase either in heterotrophic conditions in TAP medium in the dark, in mixotrophic conditions in TAP medium in low (LL) or higher light (HL), or in pure photo-autotrophic conditions in minimum (MIN) medium under HL.

mixotrophic conditions for 7 d in liquid medium (~20 population doublings) but found no significant difference between the conditions (Fig 2D). As telomeres might reach a new steady-state level with a slower kinetic, we repeated the experiment over a period of 60 d (~200 population doublings) but again did not detect changes in telomere length regardless of the growth conditions (Fig S2).

These experiments demonstrated that *C. reinhardtii* has an active telomere maintenance mechanism and that telomere length distribution is robust with regards to perturbation in metabolic regimes under a variety of standard laboratory growth conditions.

### *C. reinhardtii* reference strains show important differences in telomere length and size distributions

Even though telomere length distribution was very stable under different growth conditions for a given strain (Fig 2), we did observe a reproducible and significant difference in mean telomere length between the three laboratory reference strains CC4350+, T222+, and CC125+ by PETRA (Figs 1B and S1C) and between T222+ and S24– by TRF Southern blot (Fig 1E). We thus wondered if related *C. reinhardtii* strains displayed inter-strain differences in telomere length distribution. To test this, we took advantage of the recent sequencing of many closely related reference strains

widely used in different laboratories across the world and which display up to 2% genetic divergence (Gallaher et al, 2015). We performed TRF analysis on 12 related *C. reinhardtii* strains to characterize their telomeres (Figs 3A and S3A). Strikingly, steady-state telomere lengths were highly variable from strain to strain, ranging from 378 ± 24 bp (mean ± SD, N = 4) in CC125+ to 3.2 ± 1.1 kb (N = 3) in cw15.J14+, encompassing nearly one order of magnitude (Fig 3B). Telomere length did not correlate with genome divergence (genetically close strains are depicted with the same color), and we did not find any obvious genomic region, as described by Gallaher et al (2015), that would cosegregate with longer or shorter telomeres. In particular, neither the mating type nor the presence or absence of a cell wall correlated with telomere length variations. The average telomere length in strain cw15.J14+ was particularly striking, and we asked whether the signal could stem from internal telomere repeats. A *Bal*31 exonuclease treatment time course showed the signal decreasing in size demonstrating that this signal indeed corresponded to terminal repeats (Fig S3B, right). In addition to length variations, some strains, such as CC503+ and CC1010+, displayed multimodal telomere length distributions (Figs 3A and B, and S3A), and the multiple peaks corresponded to terminal fragments and not internal ones (Fig S3B, left).

Interestingly, the interstitial band at ~200 bp, which was present in 11 tested *C. reinhardtii* reference strains was absent

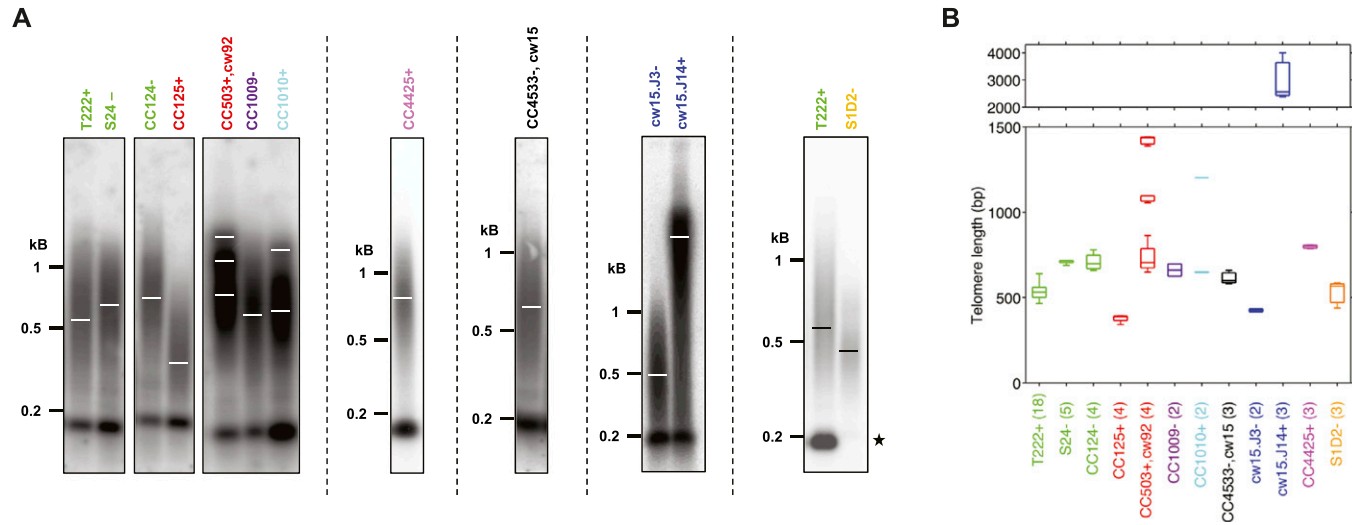

**Figure 3. Vast differences in telomere length distributions in *C. reinhardtii* reference strains.**
**(A)** Telomeres of recently sequenced *C. reinhardtii* reference strains (Gallaher et al, 2015) were analyzed by TRF analysis. Strains sharing the same name color are closely related genetically, whereas strains with different colors are more divergent. Dashed vertical lines indicate independent gels. Star: S1D2- strain does not display the band at ~200 bp. cw15 and cw92 indicate mutations that led to cell wall–less strains. **(B)** Mean and SD of telomere length for each strain as calculated by analysis of Southern blots from the indicated number of independent biological replicates (N).

from the S1D2– (CC2290–) strain (Fig 3A, star). S1D2– is an interfertile but divergent *C. reinhardtii* strain, often used for genetic mapping purposes (Gross et al, 1988; Vysotskaia et al, 2001). Thus, the interstitial telomere sequence might have emerged in a subset of *C. reinhardtii* species or conversely might have been lost in S1D2–.

## Identification of the gene encoding the catalytic subunit of telomerase

Telomerase is a holoenzyme comprising at least a reverse-transcriptase catalytic subunit and a template RNA, which are sufficient for in vitro telomerase activity (Lingner et al, 1997a). These core actors are associated with multiple other proteins, required for its recruitment, processivity, and regulation (Lewis & Wuttke, 2012). As the catalytic subunit of telomerase (e.g., hTERT in human, AtTERT in *A. thaliana*, and Est2 in *Saccharomyces cerevisiae*) is conserved, we sought to identify the gene encoding this subunit in *C. reinhardtii* and to characterize the contribution of telomerase to telomere length maintenance. Nucleotide BLAST searches in *C. reinhardtii* genome failed to find similarity to most of the shelterin or shelterin-like genes and telomerase-associated genes from human, *A. thaliana*, and *S. cerevisiae*, except for *CBF5* from *A. thaliana* (also *CBF5* in *C. reinhardtii*), corresponding to the dyskerin gene.

Gene model Cre04.g213652 of the *C. reinhardtii* nuclear genome (Phytozome v5.5; https://phytozome.jgi.doe.gov/pz/#) has a predicted N-terminal part of the corresponding protein showing partial sequence similarity with the RNA-binding domain of telomerase from a number of organisms (Fig 4A). However, the available gene model extends over 25 kb, contains 28 introns and is predicted to encode a 5,019-aa protein, much larger than telomerase from *A. thaliana* (1,123 aa), maize (1,188 aa), iris (1,295 aa), and rice (1,261 aa),

overall suggesting that the current gene model is probably incorrect. In addition, two sequencing gaps and the presence of TG and CCAC satellites both in introns and exons cloud the structure of the putative gene. Although expressed sequence tags from cDNA libraries supported the validity of some parts of the conserved 5' and 3' regions, no expressed sequence tag was found for the central part of the gene model in the available *C. reinhardtii* expression libraries. As stated above, nucleotide sequence alignments failed to detect similarity with telomerase catalytic subunit genes of other organisms. We thus performed PSI-Blast alignments of the C-terminal protein domain of the putative *C. reinhardtii* telomerase with telomerase from plants using PRALINE (http://www.ibi.vu.nl/programs/pralinewww). The alignments showed strong similarity to the C-terminal catalytic reverse-transcriptase domain of *A. thaliana* (e-value = $3 \times 10^{-36}$), maize (e-value = $4 \times 10^{-35}$), iris (e-value = $1 \times 10^{-36}$), and rice (e-value = $3 \times 10^{-24}$) (Fig 4B). The conserved C motif (mC) in organisms ranging from *S. cerevisiae* to *A. thaliana* and humans, including two critical aspartates for telomerase catalytic activity (Lingner et al, 1997b; Nakamura et al, 1997; Oguchi et al, 1999) showed strong sequence conservation with a corresponding motif in the putative *C. reinhardtii* protein (Fig 4B and C). Motif E (mE) was conserved to a lesser degree, whereas no clear conservation of motifs mA, mD, and motif 1 and 2 (Lingner et al, 1997b; Oguchi et al, 1999) was found in the predicted *C. reinhardtii* protein. Other well-conserved regions in the C-terminal part with no assigned motif are also depicted in Fig 4B.

To demonstrate that the genomic region Cre04.g213652 indeed contains the gene encoding the catalytic subunit of telomerase of *C. reinhardtii*, we selected three strains harboring insertions of the paromomycin resistance cassette within the putative gene from the recently created CliP library of mapped insertional mutants (Li et al, 2016) (https://www.chlamylibrary.org) (Fig 4A). LMJ.RY0402.077111 has an insertion in a putative intron near the region encoding the

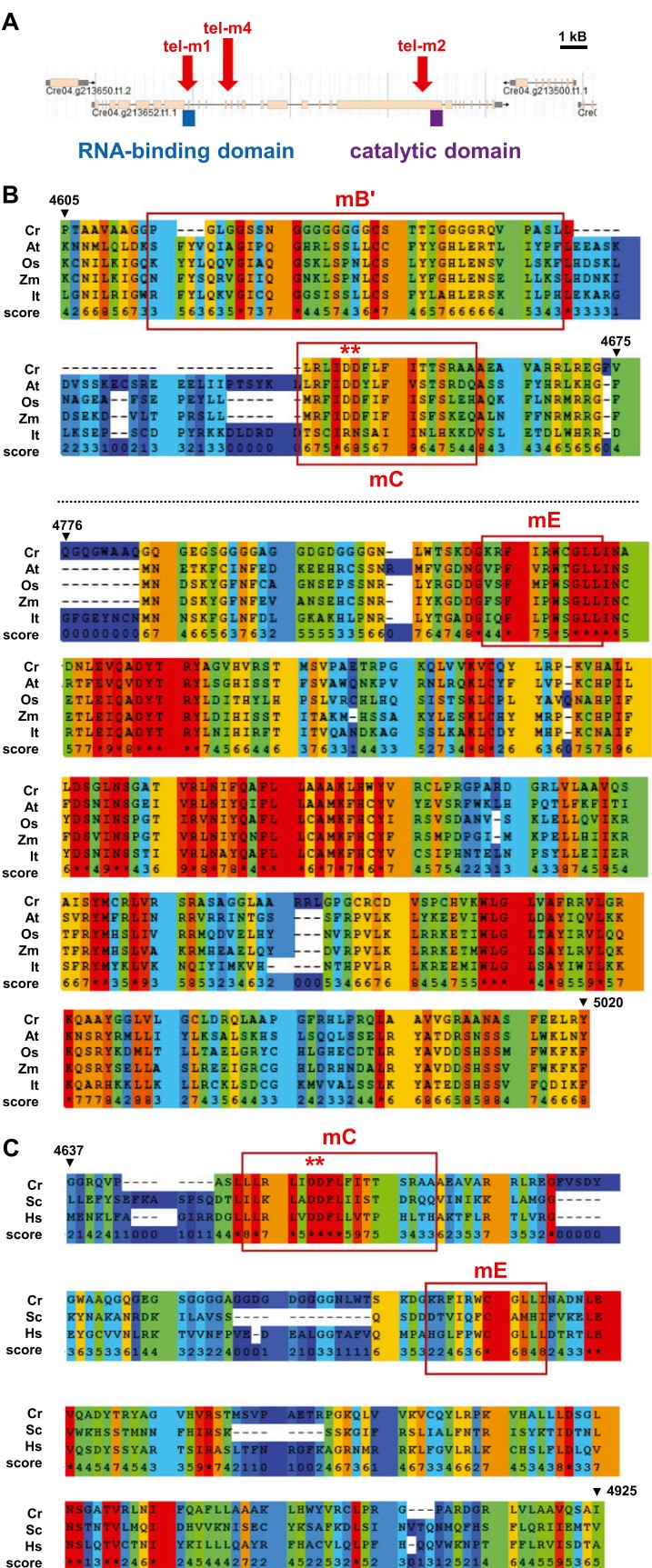

**Figure 4. Identification of the *CrTERT* gene encoding the catalytic subunit of telomerase in *C. reinhardtii*.**
**(A)** The protein corresponding to the predicted gene model Cre04.g213652.t1.1 of the available *C. reinhardtii* nuclear genome harbors an annotated N-terminal domain with significant similarities to the RNA template-binding domain of telomerases from other organisms. The C-terminal domain shows strong similarities with the catalytic domain of this enzyme in other organisms. Mutants tel-m1 (LMJ.RY0402.077111) and tel-m2 (LMJ.RY0402.209904) from the CliP library have reported insertions in either the RNA-binding or the catalytic domain, respectively. Mutant tel-m4 (LMJ.RY0402.105594) has an insertion in between these two domains. **(B)** PSI-blast alignments show strong amino-acid sequence similarity of the catalytic domain of telomerases from many organisms with the putative *C. reinhardtii* protein. Similarity score ranges from 0 (light blue) to 9 and * (red) indicates identity. The motifs B′, C, and E (mB′, mC, and mE) described by Lingner et al (1997b), Ogushi et al (1999) show strong conservation in *C. reinhardtii*, including two catalytic aspartates, essential for telomerase function in other organisms (red asterisks). Conservation can also be observed downstream of mE between *CrTERT* and the other telomerases. **(C)** The mC motif of *C. reinhardtii* shows strong sequence similarity with the mC motif containing two catalytically essential aspartates in yeast and human telomerases (Lingner et al, 1997b; Ogushi et al, 1999). Cr, *C. reinhardtii*; At, *A. thaliana*; Os, *O. sativa*; Zm, *Z. mays*; lt, *I. tectorum*; Sc, *S. cerevisiae*; Hs, *H. sapiens*.

putative RNA-binding domain of the gene and was named tel-m1. LMJ.RY0402.209904 has an insertion in the putative CDS of the putative catalytic C-terminal domain and was named tel-m2. LMJ.RY0402.105594 has an insertion in an intron in a non-conserved region between these two domains and was named tel-m4. Although the insertions in these three mutants were already mapped by the work of Li et al (2016) with a confidence of 95% for tel-m1 and tel-m4 and 73% for tel-m2, we verified that all three mutants indeed had the insertion at the predicted loci, using PCR with primers targeting the gene and/or the inserted paromomycin resistance marker (Fig S4A and B). For all three mutants, the obtained PCR products were gel-excised, sequenced, and shown to correspond to the expected genomic region. We also backcrossed mutants tel-m1 and tel-m2 with the paromomycin-sensitive T222+ strain and

analyzed the segregation of the paromomycin resistance phenotype in tetrads after germination of the diploids (Fig S4C). Paromomycin resistance systematically segregated with a 2:2 ratio in the haploid offspring, suggesting that the functional marker was not inserted at multiple loci in the genome. Correct 2:2 segregation of the mating locus in the offspring of the tetrads was checked by PCR (Fig S4D).

We then analyzed the telomere length of the three mutant strains. All three mutants showed significantly shorter telomeres when compared with the parental CC4533– strain used by Li et al (2016) to construct the CliP library (Figs 5A and S5A; mean ± SD, tel-m1: 373 ± 25 bp, N = 4, tel-m2: 383 ± 30 bp, N = 4, and tel-m4: 387 ± 12 bp, N = 2, compared with CC4533–: 614 ± 41 bp, N = 3). We verified that the shorter telomere length in mutants tel-m1, tel-m2, and tel-m4 was not simply due to the transformation protocol used to generate

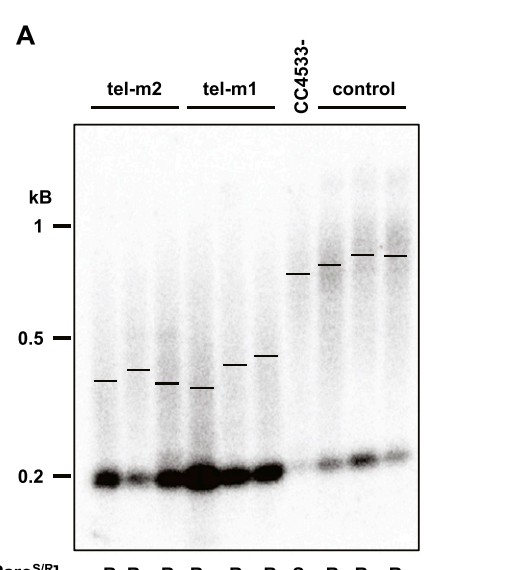

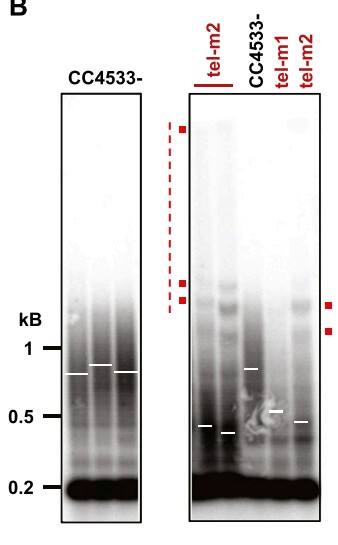

**Figure 5.  Insertional mutants of the *CrTERT* gene have shorter telomeres.**
**(A)** Mutants tel-m1 and tel-m2 have shorter telomeres in TRF analyses (three independent subclones are shown). Control: mutant LMJ.RY0402.239308 from the CliP library, which has an insertion in a gene unrelated to *CrTERT*. Paromomycin resistance phenotype is indicated ("[Paro$^{S/R}$]"; "S": sensitive, and "R": resistant). **(B)** Prolonged liquid cultures of telomerase mutants lead to rearranged TRF patterns. Cells were cultured in liquid medium for 2 mo before TRF analysis. Additional bands and slow-migrating DNA molecules (red dots and dotted vertical line, respectively) are indicated for tel-m1 and tel-m2 and are not present in the CC4533– reference strain TRF pattern. **(C)** Tetrad analysis of the cross between tel-m1 and T222+ shows a 2:2 cosegregation of paromomycin resistance and shortened telomeres after 21 and 42 d after the cross (see also Fig S5B). Mating types "mt+" and "mt–" are indicated. **(D)** Tetrad analysis of the cross between tel-m2 and T222+ shows a 2:2 cosegregation of paromomycin resistance and shortened telomeres after ~80 d after the cross (see also Fig S5C). Mating types "mt+" and "mt–" are indicated.

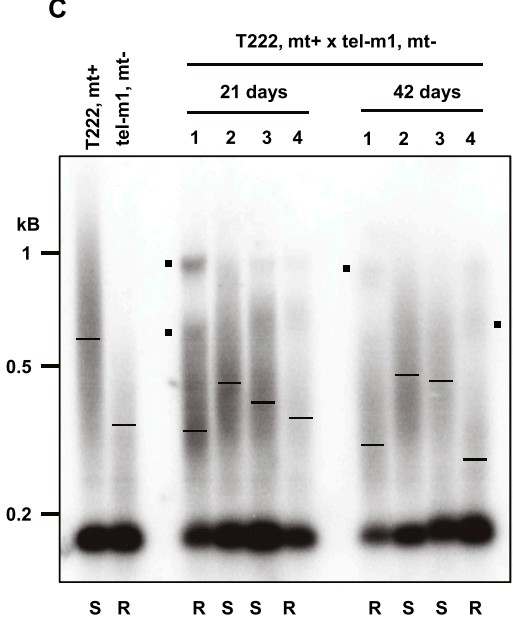

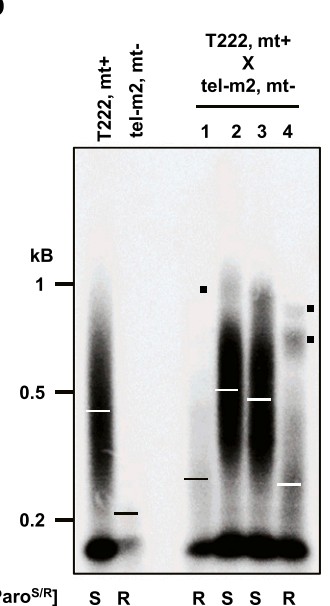

**Life Science Alliance**

the CliP library or to the insertion of the paromomycin marker itself. Telomere length was measured in another mutant from the CliP library, harboring an insertion elsewhere in the genome (on chromosome 1), and was comparable with the parental CC4533–strain (Fig 5A, "control" and Fig S5D).

We conclude that although gene model Cre04.g213652 might be wrong in its predicted structure and will require further study to be corrected, this genomic region indeed harbors the gene encoding for the catalytic subunit of telomerase in *C. reinhardtii*, and we propose to rename it *CrTERT*.

### Telomere rearrangement and maintenance in long-term cultures of telomerase mutants

Because telomeres shortened in telomerase-negative cells, we wondered whether the cells would experience replicative senescence after an extended period of growth, when telomeres reach a critically short length. We thus grew the telomerase mutants tel-m1 and tel-m2 as well as the reference strain for 2 mo (~200 population doublings), with dilutions into fresh TAP medium every 5 d. We did not observe *CrTERT* mutant cultures dying out or any obvious growth defect at any time point. However, TRF analysis of these long-term cultures of tel-m1 and tel-m2 showed alterations of their telomere length distribution (Fig 5B, compare with Fig 5A): additional discrete bands appeared at sizes above 1 kb (red dots), and a signal that extended up to the wells was detected (vertical red line). Interestingly, the three independent cultures of the tel-m2 mutant gave similar but distinct patterns with respect to the discrete bands and the high molecular weight signal. The tel-m1 mutant also displayed some additional bands, albeit not to the extent of tel-m2. Overall, these altered TRF patterns observed in prolonged cultures of telomerase mutants are reminiscent of TRF patterns observed for cells with telomerase-independent maintenance pathways (e.g., type II survivors of telomerase-negative yeast cells or alternative lengthening of telomeres (ALT)–like telomerase-negative cancer cells. See the Discussion section.)

### Telomeres shorten progressively in telomerase mutants

The initial CliP telomerase mutants might have accumulated additional, potentially suppressor, mutations, which could interfere with the proper assessment of the mutant phenotype. Importantly, the presence of suppressor mutations could explain why these mutants did not show any discernible growth defects in standard growth conditions or any sign of senescence after prolonged culture.

To outcross potential suppressor mutations and gain a kinetic perspective on telomere shortening in the telomerase mutants, we backcrossed mutants tel-m1 and tel-m2 with a wild-type strain of opposite mating type (T222+) and, after germination of the diploids, studied the telomere length distribution of the obtained tetrads. Backcrossing a mutant cell with a telomerase-positive strain should allow telomerase to elongate the shortest telomeres brought in by the mutant strain. The subsequent meiosis would then shuffle the chromosomes and the telomeres in the four haploid cells, independently of the mutant or wild-type status of the telomerase gene. We thus expect that immediately after meiosis in the diploid,

the four haploid cells would have similar and nearly wild-type average telomere length. After culture, the telomere length in the four progenies should vary according to the status of the *CrTERT* gene.

Strikingly, measurement of telomere length in the four haploid progenies of the tel-m1– x T222+ cross after 21 d, the earliest time point we could obtain, showed that two of them displayed on average longer telomeres and the other two shorter telomeres, which corresponded to the telomerase mutants as assessed by paromomycin resistance (Fig 5C, "21 d"). After 21 more days, the telomeres of the telomerase-positive cultures maintained or increased their average length, whereas the telomerase-negative cultures displayed further shortening of their telomeres (Fig 5C, "42 d," and Fig S5B). A similar result was observed for the progenies of the cross tel-m2– x T222+ (Figs 5D and S5C). Telomere length could not be assessed in the dormant diploid state, as diploid cells could not be grown. These results strongly argued against the possibility that the shorter telomeres observed in tel-m1 and tel-m2 were due to additional mutations in the genome because they would not necessarily have co-segregated with the paromomycin marker. We also noted the presence of other bands and peaks in the smear, which were likely the result of segregating parental telomeres of very different lengths during meiosis (black dots in Figs 5C and D, and S5B).

Although no growth defect was observed for the initial tel-m1 and tel-m2 mutants, analysis of the haploid progeny from backcrosses between tel-m1 and tel-m2 with the wild-type T222+ strain (n = 4 independent tetrads, with eight telomerase-negative haploids) showed that four of the eight telomerase-negative haploid progenies experienced growth defects and then massive cell death, typical of replicative senescence (highlighted in red in the table of Fig S5E). Strikingly, for each of these four telomerase-negative progenies that experienced massive cell death, some cells managed to form colonies again at very low frequency (Fig S5E, left) and thus corresponded to postsenescence survivors. The four other telomerase-negative haploid progenies did not display any growth defect (highlighted in green in the table of Fig S5E). Individual colonies of postsenescent survivors kept on solid media showed cycles of moderate growth and subsequent cell death. This complex and dynamic survivor phenotype will be investigated in future studies.

## Discussion

In this study, we provide a detailed molecular characterization of *C. reinhardtii* telomeres by investigating their sequence, end structure, and length distribution. We also identify *CrTERT*, the gene encoding the catalytic subunit of telomerase, and find that mutants of this gene experience telomere shortening and can enter replicative senescence.

### Telomere sequence and end structure

Because telomere-bound proteins specifically interact with telomere sequences (Palm & de Lange, 2008; Fulcher et al, 2014), the

variability of telomeric repeat motif can have functional consequences (Arneric & Lingner, 2007; Marzec et al, 2015). *C. reinhardtii* telomeric repeats are mostly nondegenerate, with few low-frequency variants, notably repeats of the canonical *A. thaliana* type (TTTAGGG), possibly as a remnant of the ancestral motif in the green lineage (Fulnečková et al, 2012). The overall low occurrence of variants (Table 1) suggests that *C. reinhardtii* telomerase is a high-fidelity reverse transcriptase, in contrast to telomerase from other unicellular eukaryotes such as *S. pombe* or *S. cerevisiae* (Zakian, 1995).

Analysis of the available genome sequences of *C. reinhardtii* strains and our TRF analysis suggests the presence of interstitial telomeric repeats, which might be due to chromosome end-to-end fusion over the course of evolution (Meyne et al, 1990; Azzalin et al, 2001; Uchida et al, 2002; Gaspin et al, 2010; Aksenova et al, 2015). But whether they might have any functional role, for example, as binding site for transcription factors (Platt et al, 2013) remains to be investigated.

The protective function of telomeres is not solely provided by their sequence and associated proteins but also by the structure formed at the termini. Similar to *A. thaliana*, *C. reinhardtii* telomeres seem to comprise two subsets with different end structures: 3′ overhangs and blunt ends. The latter might correspond to telomeres replicated by the leading strand synthesis, which naturally generates a blunt end. Although we cannot exclude that the structures we detect might represent transient blunt ends that then undergo further processing, the very short relative duration of the S phase in *C. reinhardtii* cell cycle rather suggests that the blunt ends are stable structures. Although 3′ overhangs are common to many eukaryotic species, blunt ends have been observed in plants, mostly in angiosperms but not in the moss *Physcomitrella patens* (Kazda et al, 2012). Our discovery that the blunt ends are present in green algae suggests that this structure is of much older evolutionary origins than previously thought.

### Intra-strain stability and inter-strain variations in telomere length distribution

Telomere length is regulated by multiple pathways, as shown by exhaustive screens performed in *S. cerevisiae* (Askree et al, 2004; Gatbonton et al, 2006; Ungar et al, 2009; Chang et al, 2011), including nucleic acid metabolism, DNA replication, chromatin modification, and protein degradation, among others. In addition, telomere length is also sensitive to both internal and environmental cues (Walmsley & Petes, 1985; von Zglinicki, 2000; Epel et al, 2004; Cetin & Cleveland, 2010; Romano et al, 2013; Fulcher et al, 2014; Millet et al, 2015; Millet & Makovets, 2016). We found no change in telomere length distribution when *C. reinhardtii* cells were grown in a wide variety of physiologically relevant laboratory conditions, including growth phases, carbon sources, and light conditions. Although we cannot exclude that other harsher growth conditions or internal signaling (e.g., DNA damage or replication stress) might induce an alteration in telomere length or structure, this result suggests that the mechanisms maintaining telomere length homeostasis are highly robust and efficient.

In stark contrast, closely related strains of *C. reinhardtii* displayed very different telomere length profiles, similar to variations observed in different strains, isolates, or ecotypes of other species (Walmsley & Petes, 1985; Burr et al, 1992; Zhu et al, 1998; Shakirov & Shippen, 2004; Raices et al, 2005; Liti et al, 2009; Fulcher et al, 2015) and suggesting that a complex network of genetic regulation controls telomere length. A detailed functional genetic approach to map the regions of the genome responsible for telomere length variation could identify pathways regulating telomere length.

Overall, the diversity of telomere length distributions observed in these reference strains highlights the plasticity of telomere length regulation and the phenotypic heterogeneity of *C. reinhardtii* reference strains.

### Identification of *CrTERT* encoding the catalytic subunit of telomerase

Sequence similarity and functional analyses of three independent mutant alleles suggest that the gene model Cre04.g213652 corresponds to, or at least encompasses, the gene encoding the catalytic subunit of telomerase, required to maintain telomere length in *C. reinhardtii*. We propose to rename it *CrTERT*.

Multiple lines of evidence support this conclusion. First, the predicted protein shares significant sequence similarity with the RNA-binding domain of telomerase from other organisms in its N terminus. Second, we find a very strong conservation of the C-terminal domain of the proposed CrTERT protein, including two essential aspartates, with the catalytic domain of telomerase not only from plants (maize, *A. thaliana*, soya, and iris) but also from yeast and human. Third, three independent mutants (tel-m1, tel-m2, and tel-m4) with different insertions of the paromomycin resistance marker in *CrTERT*, including within its RNA-binding domain (tel-m1) and its catalytic domain (tel-m2) display significantly shorter telomeres than the parental CC4533− strain, which is not the case for other independent mutants from the CLiP library located in loci unrelated to telomerase. Finally, telomeres shortened progressively in paromomycin-resistant progenies. However, as we are as of yet unable to detect the mRNA corresponding to *CrTERT* by either Northern blotting or RT-qPCR (Reverse Transcription Quantitative PCR), possibly because of its low expression, we could not assess *CrTERT* expression in our study. The identification of additional components of the telomerase holoenzyme and telomere-associated proteins will be the focus of future work.

### Telomere shortening, replicative senescence, and alternative maintenance pathways

After prolonged liquid cultures of multiple independent tel-m1 and tel-m2 mutants, we observed a drastically altered TRF pattern: discrete bands above 1 kb and a continuous smear of high molecular weight fragments up to the wells. These new TRF signals could correspond to extremely long telomeres, as seen for strain cw15.J14+ and also to DNA molecules with abnormal structures, such as G-quartets, other secondary structures, or single-stranded DNA. These rearrangements might be produced by alternative mechanisms of telomere maintenance or elongation and are reminiscent of telomere profiles observed in type II postsenescent yeast cells (Lundblad & Blackburn, 1993), ALT cancer cells (Cesare & Reddel, 2010; Shay et al, 2012), or ALT *A. thaliana* cell lines (Zellinger et al,

2007; Akimcheva et al, 2008), in which telomerase-independent recombination mechanisms can lead to very long and heterogeneous telomeres, thus sustaining long-term cell divisions. In these described cases, telomerase is not expressed, telomeres undergo sister-chromatid, and interchromosome homologous recombination using gene conversion, break-induced replication, rolling circle amplification, or yet unknown mechanisms.

Another line of evidence suggesting the occurrence of post-senescence survivors of telomerase-negative cells in *C. reinhardtii* came from the analysis of the offspring of backcrosses of the tel-m1 and tel-m2 mutants with T222+ reference strain. The *CrTERT*-mutant progenies experience telomere shortening, and 50% of them eventually stopped growing after about 6 mo on solid media, a phenotype consistent with replicative senescence. The other 50% of telomerase-negative progenies have not yet entered senescence at the date of publication of this work (>21 mo). The cells that experienced senescence and generated first generation survivors then showed a complex pattern of moderate growth, followed by cell death and emergence of a new generation of clonal survivors. We do not yet understand the dynamics and variability of the senescence phenotype in these backcrossed haploid progenies. We speculate that for the initial CliP mutants, additional mutations could have been generated that might have acted as suppressors of the senescence phenotype. This would also explain why no growth defect was observed for the initial CliP mutants even after more than 2 yr of maintenance on solid media, whereas senescence, cell death, and postsenescent survivors could be observed after backcrossing these mutants and selecting telomerase-negative haploid progenies. Alternatively, the initial CliP mutants might have already been postsenescence survivors from the beginning. In a future work, it will be interesting to characterize postsenescence survivors by assessing hallmarks of human ALT cancers, including circular extrachromosomal telomeric DNA and up-regulation of telomeric repeat–containing RNA (TERRA) (Cesare & Reddel, 2010; Arora & Azzalin, 2015).

Although some fundamental aspects of its telomeres share similarities to other eukaryotes, *C. reinhardtii* shows a unique combination of telomeric properties that distinguishes it from any other model organism. The characterization of its telomeres at the level of sequence, end structure, length distribution, and maintenance by telomerase or alternative mechanisms provided by this study is an essential step to propose *C. reinhardtii* as a valuable model organism for telomere biology research.

# Materials and Methods

### Strains and growth conditions

Strains T222+, S24−, CC124−, CC125+, CC503+, CC1009−, CC1010+, and CC4425+ (D66) are described in Gallaher et al (2015). Strains cw15.J3− and cw.J14− are cell wall–less strains obtained by crossing. Strains CC620+, CC521+, and CC4350+ are described in the Chlamydomonas Resource Center (https://www.chlamycollection.org/). Strain CC4533− is described in the work by Li et al (2016). Strain S1D2 is described in the work by Gross et al (1988), Harris (2001), and Vysotskaia et al (2001). Unless stated otherwise, the cells were grown under continuous illumination either on plates or in agitated 200-ml liquid cultures in TAP medium (Harris, 2009) under low-light, that is, 8 $\mu$E·m$^{-2}$·s$^{-1}$ or higher light (HL), that is, 80 $\mu$E·m$^{-2}$·s$^{-1}$. The *A. thaliana* ecotype Columbia (Col-0) plant was used for the hairpin assay.

### gDNA extraction

Unless stated otherwise, the cells were grown in liquid cultures to early stationary phase (~2 × 10$^7$ cells·mL$^{-1}$) and 150 ml was collected by centrifugation (5,000$g$, 5 min). The pellet was frozen at −80°C. The cells were then thawed at room temperature and 5 ml of preheated buffer AP1 with RNase (QIAGEN DNA Plant Maxi Kit) was added and cells lysed at 65°C for 2 h. After lysis, gDNA was extracted according to the manufacturer's protocol (QIAGEN DNA Plant Maxi Kit). For PETRA and hairpin assays, gDNA was extracted using the CTAB method as described in Borevitz et al (2003).

### Telomere PCR and sequencing

Bulk gDNA was denatured at 95°C during 5 min. End labeling reactions (total volume 6 $\mu$l) contained 100 ng of bulk gDNA, 1× New England Biolabs restriction buffer 4, dCTP 100 $\mu$M, and 1 unit of terminal transferase (New England Biolabs, NEB) and was carried out at 37°C during 30 min, then 65°C during 10 min, and 94°C during 5 min. The end-labeled telomeres were then amplified with the primers 169M (poly-G–containing primer) and oT1090 targeting the subtelomere/telomere junction common to 10 telomeres of eight chromosomes (Table S1). PCR reactions (40 $\mu$l) contained the end-labeled DNA, 200 $\mu$M of dNTPs, primers at 0.5 $\mu$M for oT1090 and 0.75 $\mu$M for 169M, 1× Taq Mg-free buffer (NEB), and 2.5 U of standard Taq polymerase (NEB). The PCR conditions were as follows: 94°C 3 min; 32 cycles of 94°C 20 s, 60°C 40 s, and 68°C 20 s; and 68°C 5 min.

For sequencing, PCR products were ligated for 1 h at 16°C in a pDrive plasmid. 2 $\mu$l of the ligation product was transformed into competent bacteria (PCR cloning kit; QIAGEN). Bacteria were plated on LB + ampicillin (100 $\mu$g/ml) + IPTG (50 $\mu$M) + X-gal (80 $\mu$g/ml) medium overnight at 37°C. Plasmids were extracted and purified (Millipore Plasmid Miniprep 96 Kit and Manifold) after 24 h of culture of white colonies in 1 ml of LB 2X + ampicillin (100 $\mu$g/ml) in a 96-well microplate. DNA insertion in plasmids was verified by *Eco*RI (NEB) digestion. Plasmids were Sanger-sequenced with the M13-PU primer (Eurofins Genomics).

### Isolation of nuclei

Nuclear fraction was prepared from cell wall–mutant CC4350+. Exponentially growing cells (2 d in liquid culture) were gently spun and thoroughly resuspended in 90 ml buffer A per liter of culture (25 mM Hepes–NaOH, pH 7.5, 20 mM KCl, 20 mM MgCl$_2$, 600 mM sucrose, 10% glycerol, and 5 mM DTT). Triton X-100 was first diluted in 10 ml of buffer A per liter of culture and subsequently added drop wise to the cells while swirling them gently, to a final concentration of 0.5%. Nuclei were pelleted at 800 g for 2–4 min. Using a paintbrush, the pellet was gently resuspended in fresh buffer A without Triton X-100. After centrifugation, the integrity of nuclei (1–5 $\mu$l) was checked by fluorescent microscopy using DAPI/vectashield (5 $\mu$l)

staining. Nuclei were resuspended in buffer B (2.5% Ficoll 400, 0.5 M sorbitol, 0.008% spermidine, 50% glycerol, and 1 mM DTT), using 1–2 ml per 200 ml of original culture volume. For storage, nuclei were frozen in liquid nitrogen and stored at –80°C.

### MNase hypersensitivity assay

MNase hypersensitivity assay was based on Lodha & Schroda (2005). 1 ml of nuclei isolated from *C. reinhardtii* strain CC4350+ cw15 mt+ in buffer B was thawed on ice. To collect nuclei, the sample was spun at maximal speed for 15 s and resuspended in 500 $\mu$l of 1× MN buffer (50 mM Tris–HCl, pH 8.0, and 5 mM CaCl$_2$). Reactions of total volume of 110 $\mu$l were carried out in 1× MN buffer using 60 $\mu$l of sample and different amounts of MNase units (Fermentas). gDNA from *C. reinhardtii* CC4350+ (~750 ng) was used as a control for enzyme activity and digested with 15 U of nuclease. The samples were incubated 3 min at room temperature, and reactions were stopped by adding 110 $\mu$l of STOP buffer (1% SDS and 50 mM EDTA). Proteins were then denatured for 45 min at 65°C and DNA was extracted using 500 $\mu$l of phenol:chloroform:isoamyl alcohol (25:24:1) using Phase Trap A (Peqlab). Aqueous phase was precipitated by adding 42 $\mu$l of 3 M NaOAc and 840 $\mu$l of 96% ethanol. DNA was pelleted by centrifugation for 10 min at maximal speed. The pellet was washed with 70% ethanol, dried, and resuspended in 25 $\mu$l H$_2$O. 6× loading dye (6 $\mu$l) was added prior loading onto 1.5% agarose gel. DNA was stained with ethidium bromide (1% solution; AppliChem), blotted onto uncharged membrane (Amersham), and hybridized with a (T$_4$AG$_3$)$_3$ probe. After scanning, the membrane was stripped and reprobed using 18S-derived probe.

### PETRA and hairpin assay

Telomere length of individual chromosomes was determined by Primer Extension Telomere Repeat Amplification (PETRA) as previously described (Watson et al, 2016). For primer extension by the Φ29 polymerase, we used the *C. reinhardtii*–specific PETRA-T oligonucleotide 5′-CTCTAGACTGTGAGACTTGGACTACCCTAAAACCCT-3′ (Table S1). For specific chromosome arms, we used subtelomeric oligonucleotides 1R: 5′-TACTTGTGTGTGCTGTGCGT-3′, 9R: 5′-ACAG-CACAATACAGTATATA-3′, and 10R: 5′-AACGTCCTCGTGAGACCACC-3′ (Table S1). The hairpin assay for detecting blunt-ended telomeres was performed as previously described (Kazda et al, 2012). Southern hybridization was performed with a [$^{32}$P]ATP-labeled (TTTTAGGG)$_3$ probe. PETRA membrane was also hybridized with [$^{32}$P]ATP-labeled 1-kb ladder (Thermo Fisher Scientific).

### TRF and in-gel hybridization analyses

2 $\mu$g of gDNA was digested in 300 $\mu$l with a cocktail of six restriction enzymes (*Pst*I, *Bam*HI, *Mnl*I, *Fok*I, *Taq*I, and *Msp*I; 20 units each). Digestion products were isopropanol precipitated, resuspended in loading buffer (gel loading dye, Purple 6X, New England Biolabs) and resolved on a 1.5% agarose gel for 4 h at 150 V. The gel was then soaked in a denaturation bath (0.4 M NaOH and 1 M NaCl) for 20 min and transferred overnight by capillary to a charged nylon membrane (Hybond XL; GE Healthcare). The CHSB *Chlamydomonas* telomere-specific oligonucleotide probe (Fulneckova et al, 2013)

(5′-GTTTTAGGGTTTTAGGGTTTTAGGGTTTTAG-3′, Table S1) was $^{32}$P-labeled at the 5′ terminus with ATP ($\gamma$-$^{32}$P) by the T4 polynucleotide kinase (New England Biolabs). The membrane was hybridized using the Rapid-hyb Buffer protocol (GE Healthcare). In brief, the membrane was prehybridized at 42°C in Rapid-hyb buffer for 1 h, then the radioactive probe (20 pmol) was added, and the incubation was continued for 1 h. The membrane was washed consecutively with 5× SSC, 0.5% SDS (42°C for 10 min); 5× SSC, 0.1% SDS (42°C for 20 min); and 1× SSC, 0.1% SDS (25°C for 30 min). A phosphor screen was exposed to the membrane and imaged with a Typhoon FLA 9500 scanner (GE Healthcare). Average telomere length was assessed using ImageJ 1.49v (NIH) by measuring the peak of the telomere length distribution signal. For multimodal telomere length profiles, the multiple peaks were measured. We used TeloTool, a software for TRF analysis with a built-in probe intensity correction algorithm (Gohring et al, 2014), to verify that unequal telomeric probe binding was negligible in our conditions. For in-gel hybridization analysis, gDNA was digested by the cocktail of enzymes following the same procedure, with some samples being pretreated with *Exo*T (50 units for 2 h at 25°C; New England Biolabs), which degrades single-stranded 3′ DNA extension and generates blunt ends, similarly to *Exo*I. Samples were then run in a 1× Trisborate EDTA (TBE) 0.75% agarose gel in 1× TBE buffer for 18 h at 20 V. The gel was then dried and hybridized overnight at 37°C with radioactively labeled probes (oT0958, G-probe and oT0959, C-probe, Table S1) in hybridization buffer (5× SSC, 5 $\mu$M inorganic pyrophosphate, 1 mM Na$_2$HPO$_4$, 5× Denhardt's solution, 40 nM ATP, and 20 $\mu$g/ml salmon sperm DNA). The gel was then washed three times for 30 min at room temperature with 0.25× SSC and imaged as for Southern blots. For loading controls, the gel was then transferred in denaturing conditions on a charged nylon membrane and hybridized again with the same probes.

## Supplementary Information

## Acknowledgements

We thank Erin Henninger for her help in setting up the in-gel experiment. We thank the Chlamydomonas Mutant Library Group at Princeton University, the Carnegie Institution for Science, and the Chlamydomonas Resource Center at the University of Minnesota for providing the indexed Chlamydomonas insertional mutants. This work was supported by the Agence Nationale pour la Recherche (ANR) grant "AlgaTelo" (ANR-17-CE20-0002-01) to Z Xu, la Fondation de la Recherche Médicale (MTT "équipe labellisée") and the ANR grant "InTelo" (ANR-16-CE12-0026) to MT Teixeira, the "Initiative d'Excellence" program from the French State (Grant "DYNAMO," ANR-11-LABX-0011), and by the Ministry of Education, Youth and Sports of the Czech Republic, European Regional Development Fund-Project "REMAP" (No. CZ.02.1.01/0.0/0.0/15_003/0000479) to K Riha.

### Author Contributions

S Eberhard: conceptualization, formal analysis, supervision, investigation, methodology, and writing—original draft, review, and editing.

S Valuchova: conceptualization, formal analysis, investigation, methodology, and writing—review and editing.

J Ravat: formal analysis, investigation, and writing—review and editing.

J Fulneček: formal analysis, investigation, methodology, and writing—review and editing.

P Jolivet: formal analysis, investigation, methodology, and writing—review and editing.

S Bujaldon: formal analysis, investigation, and writing—review and editing.

SD Lemaire: formal analysis and writing—original draft, review, and editing.

F-A Wollman: conceptualization, formal analysis, and writing—review and editing.

MT Teixeira: conceptualization, formal analysis, and writing—review and editing.

K Riha: conceptualization, formal analysis, supervision, and writing—review and editing.

Z Xu: conceptualization, formal analysis, supervision, investigation, methodology, and writing—original draft, review, and editing.

## Conflict of Interest Statement

The authors declare that they have no conflict of interest.

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
