## [Reviewer comments · Life Science Alliance]

Life Science Alliance

Molecular characterization of *Chlamydomonas reinhardtii* telomeres and telomerase mutants

Stephan Eberhard, Sona Valuchova, Julie Ravat, Jaroslav Fulneček, Pascale Jolivet, Sandrine Bujaldon, Stéphane Lemaire, Francis-Andre Wollman, Maria Teixeira, Karel Riha, and Zhou Xu
DOI: <https://doi.org/10.26508/lsa.201900315>

Corresponding author(s): Zhou Xu, CNRS - Sorbonne Université and Stephan Eberhard, Sorbonne Université - CNRS;

Review Timeline:

Submission Date:	2019-01-23
Editorial Decision:	2019-02-13
Revision Received:	2019-05-13
Editorial Decision:	2019-05-21
Revision Received:	2019-05-27
Accepted:	2019-05-27

Scientific Editor: Andrea Leibfried

Transaction Report:

February 13, 2019

Re: Life Science Alliance manuscript #LSA-2019-00315-T

Dr. Zhou Xu
CNRS - Sorbonne Université
Institut de Biologie Paris Seine, Laboratoire de Biologie Computationnelle et Quantitative -
UMR7238
4 place Jussieu
75252 Paris
France

Dear Dr. Xu,

Thank you for submitting your manuscript entitled "Molecular characterization of *Chlamydomonas reinhardtii* telomeres and telomerase mutants" to Life Science Alliance. The manuscript was assessed by expert reviewers, whose comments are appended to this letter.

As you will see, the reviewers think that your work is not providing new insight into telomere biology while likely being of interest to the *C. reinhardtii* community. The reviewers also raise a few technical concerns. Considering that *C. reinhardtii* is an important model organism, we decided to invite you to submit a revised version of your work, addressing the issues raised by the reviewers. Importantly, the requested controls need to get performed and included. While adding TRAP assays and analysing catalytic dead RT mutants would be a nice addition to this body of work, including such analyses is not mandatorily needed for acceptance here.

Thank you for this interesting contribution to Life Science Alliance. We are looking forward to

receiving your revised manuscript.

Sincerely,

B. MANUSCRIPT ORGANIZATION AND FORMATTING:

Reviewer #1 (Comments to the Authors (Required)):

In this manuscript, Eberhard et al characterize several properties of the telomeres of *Chlamydomonas reinhardtii*, a photosynthetic green alga. This organism is not only prominent in the ecological and biotechnology arenas, but also can provide a valuable evolutionary perspective on telomere maintenance and regulation. The authors find that *C. reinhardtii* harbors almost exclusively nondegenerate telomeres whose lengths remain constant from log to stationary phase and under varying growth conditions. Intriguingly, however, telomere length varies dramatically between different reference strains that show up to 2% genetic divergence, although the degree of telomere length difference does not correlate with degree of genetic divergence. The authors identify the likely gene encoding the catalytic subunit of the telomerase reverse transcriptase, and show that normal telomere length regulation depends on this gene.

The paper provides a useful and thorough framework for considering the basis for similarities and differences between telomere maintenance and regulation in *C. reinhardtii* and other species, and the experiments are generally very well performed. However, several issues need to be addressed before publication:

- The EtBr stained gel in Figure 1A shows a rather diffuse pattern without the regular nucleosome arrays seen for bulk chromatin in other organisms. Is this expected for *C. reinhardtii*? More likely the chromatin preparation has been somewhat degraded, which would make it difficult to evaluate the smeary and faint pattern seen for the telomere. It may also be that the telomere micrococcal pattern is more volatile than nucleosome arrays elsewhere in the genome. If the diffuse pattern holds up for the *C. reinhardtii* telomeres, it still cannot be referred to as nonnucleosomal - there may be nucleosomes with noncanonical protection properties, as found in fission yeast (Greenwood et al 2018), unless proven otherwise.
- The protein Gbp1 is mentioned as a ss telomere binding protein, but no description is given of the properties of this protein or its significance. Does it contain OB folds or other domains homologous to those of any known telomere proteins? What is known about its function in general?
- Likewise, this paper should outline the predicted presence or absence of the shelterin components. Is there a TRF-type gene in the genome, is there a Rap1, a Pot1 and so on? In this introductory paper about *C. reinhardtii* telomeres such questions will be on the minds of readers. The authors do not need to characterize all such proteins, but need to do the homology searches and discuss this.
- In Figure 1B, the PETRA assay needs a control in which the sample is treated with *E. coli* ExoI beforehand to verify that ss overhangs explain the results. This is especially important since native gels are not shown, so this PETRA result is the sole evidence for 3' overhangs.
- The hairpin assay also needs to include positive controls - *Arabidopsis* DNA would be ideal, and they also need a sample in which the ends are blunted, eg with Klenow. Otherwise, it is not clear whether blunt ends would be detected by this assay, so the negative data is not conclusive regarding their absence.
- The authors draw lines indicating average telomere lengths in several figures, but do not describe how these averages were determined. Did they take into consideration the fact that longer telomeres will hybridize more intensely to a telomere probe and therefore will appear overrepresented in the distribution, unless corrected for in determining the average?
- The putative CrTERT gene has 28 introns - can the authors comment on whether this is a high or normal intron number for genes in this organism?
- The homologies found in CrTERT with RNA binding domains and reverse transcriptase domains from other organisms along with the telomere shortening phenotype seen in the disruption strain are indeed suggestive that this gene is CrTERT, but it's not out of the question that this gene is a telomere length regulator rather than the enzyme itself, or that there are multiple telomerase encoding genes. It would be ideal for the authors to perform TRAP assays on the wild type and

disruption strains, as would mutation of the putative RT catalytic residues.

- The authors do not mention whether mutations can be readily introduced in this organism, so this reviewer cannot be sure whether the latter experiment is possible. If not, could wt and RT-mutant alleles be introduced into the tel-m1/2 mutant cells? Please include in the Introduction a description of what genetic tools are available in *C. reinhardtii*, as well as basics like the number of chromosomes.
- Fig 5B is difficult to evaluate because of the high level of background hybridization - is there a high level of DSBs or are there dying cells, or can a new gel be provided?
- The definition of 'mt' in Figs 5C and 5D is hard to find, if anywhere.
- Assuming that CrTERT is indeed the sole telomerase RT, it appears that survivors of telomerase deletion arise readily, which is interesting. The experiments in figure 5C/D in which cells with and without telomerase are mated and telomere length in the offspring followed with time are useful, but lack crucial information about telomere length in the diploid. If the authors could show that telomere length is restored in the diploid, and erodes over time (more time points than just 21 days and 42 days), an 'ever shorter telomeres' phenotype could be nicely documented; the two time points shown do not outline an EST phenotype. This would shore up the idea that CrTERT is telomerase. Does the CrTERT gene vary at all between isolates (as determined by PCR and sequencing) that have varying telomere length setpoints?
- In Figure S4, please label the diagrams on the left with numbers of bp. Ideally a long-template polymerase would be used to confirm the predicted long products in the 'no amplification' lanes.
- Check for typos and grammatical errors. The telomeres should be referred to as 'degenerate', not 'degenerated' - those words have different meanings. 'Discreet' should be spelled 'discrete' for this meaning. Some lines, like line 351, should be checked for clarity.
- A Discussion that places the results in an evolutionary context more clearly would enhance the interest of the paper, for instance discussing the variability in using telomerase as a length regulator across eukaryotes. Is telomere length variable among different isolates in other species so far studied? Do the authors propose a biological rationale for high variability in *C. reinhardtii*?

Reviewer #2 (Comments to the Authors (Required)):

The manuscript by Eberhard and colleagues sets the stage for a better understanding of telomere biology in the alga *C. reinhardtii*. The authors show that: 1) telomeres comprise mostly non degenerate T4AG3 repeats, which confirms previous work; 2) telomeres are likely not wrapped around canonical nucleosomes; 3) telomeres end with a G-rich overhang; 4) telomere length within the same strain is not affected by culture conditions, while it varies substantially amongst different strains; 5) telomerase maintains telomere length in the alga, and telomerase ablation leads to telomere shortening followed by senescence and possible appearance of ALT-type survivor cells. This study is interesting, and although it does not present any major conceptual advance in telomere biology, the study of this specific model organism is highly relevant. The data are well presented and experiments are sound. The manuscript is well written. While in principle I support its publication, a few issues should be considered and fixed.

1) The PETRA experiments need to be better controlled, as a minimum genomic DNA samples pre-treated with ExoI should be included. Also, while PETRA represent a necessary choice for telomeres with very short overhangs, there is no evidence that this is the case in *C. reinhardtii*. Native TRF in gel hybridization should be performed as a less indirect way to detect overhangs. Also here, ExoI controls should be included, and possibly parallel hybridizations with G-rich telomeric probes should be performed to test for the existence of C-overhangs, which exists in other species.

2) The 200 bp intrachromosomal telomeric repeat band seems to be sensitive (albeit less than

telomeres) to Bal31 treatment, contrarily to what the authors state.

3) In Figure 3B, I am not convinced that the shorter telomere length appearing at day 8 is representing telomere dynamics; it could simply be a partial degradation of that specific DNA sample. How reproducible is this fluctuation in length? Can the authors show also a shorter exposure or less contrasted image in order to better appreciate the integrity of the 200 bp band?

4) In Figure S3B, something went wrong with lane 2 from the left. Is there much less DNA loaded? Also here it seems that DNA degradation might confound the results. Is there a minuscule asterisk pointing to that lane (I don't see any explanation in the legend)?

5) The discussion is extremely long and would profit from shortening, to assure that important messages of the study are not diluted away. Also in the results sessions several details (how Bal31 or TRF work, for example) could be avoided.

Reviewer #3 (Comments to the Authors (Required)):

This very careful researched study investigated several key components of telomere biology in *Chlamydomonas reinhardtii*, a photosynthetic unicellular green alga. Telomere sequence and structure are analyzed in detail, which confirmed the TTTTAGGG repeat sequence identified previously by Berman and colleagues, and also showed (as in the case for most species) that chromosome termini end with a G-rich overhang. Data is also provided that shows that there is significant variation in steady-state telomere length among different *C. reinhardtii* strains; although the extent of variation is striking, the result itself is not novel, given the dramatic length differences among (for example) closely related inter-fertile species of mice.

This structural analysis was combined with genetic analysis of three pre-existing CrTERT insertional mutations, thereby confirming the prior identification of the genomic locus of this gene. The resulting telomerase-defective strains exhibited the predicted progressive telomere shortening, as well as a possible survivor phenotype, although the latter was not investigated. Interestingly, crosses between telomerase-null and telomerase-proficient strain, following by meiosis to generate haploid derivatives, resulted in a telomere-length segregation phenotype, indicating that telomerase levels in the heterozygous diploid were not sufficient to re-set telomere length back to wild type levels.

Although the experiments were carefully conducted, the conclusions are largely confirmatory, with no new insights about telomere biology, either in general or in *C. reinhardtii*.

Reviewer #1 (Comments to the Authors (Required)):

In this manuscript, Eberhard et al characterize several properties of the telomeres of *Chlamydomonas reinhardtii*, a photosynthetic green alga. This organism is not only prominent in the ecological and biotechnology arenas, but also can provide a valuable evolutionary perspective on telomere maintenance and regulation. The authors find that *C. reinhardtii* harbors almost exclusively nondegenerate telomeres whose lengths remain constant from log to stationary phase and under varying growth conditions. Intriguingly, however, telomere length varies dramatically between different reference strains that show up to 2% genetic divergence, although the degree of telomere length difference does not correlate with degree of genetic divergence. The authors identify the likely gene encoding the catalytic subunit of the telomerase reverse transcriptase, and show that normal telomere length regulation depends on this gene.

The paper provides a useful and thorough framework for considering the basis for similarities and differences between telomere maintenance and regulation in *C. reinhardtii* and other species, and the experiments are generally very well performed. However, several issues need to be addressed before publication:

We thank the reviewer for the detailed and constructive comments and hope the revised version of the manuscript addresses the raised issues.

- The EtBr stained gel in Figure 1A shows a rather diffuse pattern without the regular nucleosome arrays seen for bulk chromatin in other organisms. Is this expected for *C. reinhardtii*? More likely the chromatin preparation has been somewhat degraded, which would make it difficult to evaluate the smeary and faint pattern seen for the telomere. It may also be that the telomere micrococcal pattern is more volatile than nucleosome arrays elsewhere in the genome. If the diffuse pattern holds up for the *C. reinhardtii* telomeres, it still cannot be referred to as nonnucleosomal - there may be nucleosomes with noncanonical protection properties, as found in fission yeast (Greenwood et al 2018), unless proven otherwise.

In the experiment, we have used increasing amounts of MNase. While treatment with 30 units of MNase mostly highlights the band at ~150 bp corresponding to mononucleosomes, the lane displaying the digestion with 1 unit of MNase shows a clear regular pattern in both ethidium-bromide-stained gel and 18S-rDNA-probed membrane, with higher molecular weight bands at expected sizes. We have now added asterisks to highlight the position of the bands corresponding to mono-, di- and tri-nucleosomes.

In contrast, in none of the lanes does the telomeric signal display any band at all. Besides, the telomeric signal has a higher molecular weight than the signal for 18S rDNA, indicative of protection from MNase and not degradation during preparation. We do agree with the reviewer that we have not proven that the telosome is devoid of nucleosome. We have now added the Greenwood et al. 2018 reference and modified the text to indicate that these are non-canonical MNase-resistant structures, which might or might not be nucleosomal in nature:

“This result suggests that telomeric DNA might be fully associated with and protected by a non-canonical nucleosomal structure or by other protein complexes, similar to telosomes as observed in yeasts for example (Greenwood et al., 2018; Wright et al., 1992).”

- The protein Gbp1 is mentioned as a ss telomere binding protein, but no description is given of the properties of this protein or its significance. Does it contain OB folds or other domains homologous to those of any known telomere proteins? What is known about its function in general?

Gbp1 was mostly studied *in vitro* and found to comprise two RNA binding motifs and to bind single-stranded RNA or DNA containing two or more repeats of the *Chlamydomonas* G-strand telomere sequence TTTTAGGG. The preference toward RNA or DNA depends on the concentration and the monomeric/dimeric state of the protein. However, since the Johnston et al. paper from 1999, we have found no other study of this protein and the *in vivo* role of this protein remains unexplored. We have now added more information about Gbp1 in the text, in the introduction and in the results:

“... and (iv) the Gbp1 protein binds *in vitro* to single-stranded telomere sequences through two RNA recognition motifs, with a preference for RNA when Gbp1 is monomeric and for DNA when it is dimeric (Johnston et al., 1999; Petracek et al., 1994).”

“As it was reported that the Gbp1 protein preferentially binds single-stranded *C. reinhardtii* telomeric DNA (Johnston et al., 1999), the presence of a 3' overhang would be consistent with a role of Gbp1 at telomeres, possibly protecting them from degradation and fusions similarly to telomere capping proteins in other species.”

- Likewise, this paper should outline the predicted presence or absence of the shelterin components. Is there a TRF-type gene in the genome, is there a Rap1, a Pot1 and so on? In this introductory paper about *C. reinhardtii* telomeres such questions will be on the minds of readers. The authors do not need to characterize all such proteins, but need to do the homology searches and discuss this.

We performed homology searches at the beginning of the project and were surprised by the lack of conservation of most genes involved in telomere biology, even from *A. thaliana*. More specifically, we performed nucleotide BLAST searches for all shelterin and shelterin-like as well as telomerase components from human, *A. thaliana* and *S. cerevisiae*. We found no significant homology with any of the genes, with the exception of *A. thaliana*'s dyskerin gene *CBF5*. Studying *CBF5* might be promising to better understand telomere biology in *C. reinhardtii*.

When we extend the homology search to the components of the DNA damage response in general, we do obtain more hits but this goes beyond the scope of this study. We have now added in the text, just before the paragraph about *CrTERT* in the results: “Nucleotide BLAST searches in *C. reinhardtii* genome failed to find similarity to most of the shelterin or shelterin-like genes and telomerase-associated genes from human, *A. thaliana* and *S. cerevisiae*, except for *CBF5* from *A. thaliana* (also *CBF5* in *C. reinhardtii*), corresponding to the dyskerin gene.”

- In Figure 1B, the PETRA assay needs a control in which the sample is treated with *E. coli* ExoI beforehand to verify that ss overhangs explain the results. This is especially important since native gels are not shown, so this PETRA result is the sole evidence for 3' overhangs.

We have now added PETRA experiments that include the ExoI-treated control and another negative control that lacks the first primer extension step by the phi29 polymerase. ExoI-treated sample still showed some weak signal, which was to be expected since PETRA is an amplification-based technique, but the decrease of signal upon Exo-I treatment is clear.

As suggested by reviewer #2, we now also provide native in-gel hybridization evidence for 3' overhangs. Figure 1 has been modified to include the controls for the PETRA assay and the native in-gel experiment. Figure legends and text describing the result have been modified accordingly.

- The hairpin assay also needs to include positive controls - *Arabidopsis* DNA would be ideal, and they also need a sample in which the ends are blunted, eg with Klenow. Otherwise, it is not clear whether blunt ends would be detected by this assay, so the negative data is not conclusive regarding their absence.

The hairpin assay is extremely sensitive to the quality of the extracted DNA. We realized that our initial attempts to detect blunt ends in *C. reinhardtii* strains T222+ and S24- failed because of DNA quality (see Figure A below). Therefore, we chose to prepare new DNA of much better quality from a strain that has no rigid cell wall (CC4350+) and is easier to lyse (Figure A). Using this strain, we were able to detect blunt ends as clearly as in the *Arabidopsis* sample, which we added as a positive control, following the reviewer's suggestion. An additional negative control was performed as well, which is the pretreatment with T7 exonuclease converting potential blunt ends into 3' single-stranded DNA.

These data are now added in the manuscript and several parts of the manuscript (highlighted in yellow) have been modified to report the finding that blunt ends exist in *Chlamydomonas*. We discuss the implications of the presence of blunt end for the evolution of telomere structures in the green lineage.

Figure A. Ethidium bromide stained gels showing genomic DNA extracted from independent samples of *C. reinhardtii* strains T222+, C125+ and CC4350+ (with no rigid cell wall), and *A. thaliana*, using the same extraction method. The quality of CC4350+ DNA is comparable to that of *Arabidopsis* DNA and much better than T222+ and CC125+.

- The authors draw lines indicating average telomere lengths in several figures, but do not describe how these averages were determined. Did they take into consideration the fact that longer telomeres will hybridize more intensely to a telomere probe and therefore will appear overrepresented in the distribution, unless corrected for in determining the average?

As indicated in the supplemental methods section, we assess the average telomere length by measuring the peak of the distribution based on the intensity profile, as shown in supplemental Figure S3A. For multimodal distributions, we also indicated the other peaks in addition to the main one.

To address the concern about unequal probe binding in the TRF Southern blot, we attempted to use TeloTool (Göhring et al. 2013 NAR), a software developed by co-author Karel Riha to measure telomere length with probe intensity correction. However, for the vast majority of the samples, TeloTool failed to fit the intensity profile because of the band at ~200 bp and

produced an aberrant corrected profile (see Figure B below). Cropping the image beforehand to remove that band failed as well since a significant part of the profile is then missing, which is needed for the fit by TeloTool. Nevertheless, we were able to use TeloTool in some TRFs, where the fit somehow ignored the band at ~200 bp (see Figure C). For these samples, the corrected intensity profile generated by TeloTool closely matched the original one, suggesting that unequal probe intensity is negligible. Besides, TeloTool is not able to appropriately measure telomere length in multimodal distributions. For all these reasons, we have kept the peak-based method for measuring average telomere length, but we provide more detail in the method section to address the probe intensity issue.

Figure B. Screenshot of the TeloTool interface with an example of failed analysis and aberrant correction by TeloTool. The graph represents the analysis of the first lane. The raw data and its fit are shown in black. The corrected data taking unequal probe intensity into account and its fit are shown in red. Because the algorithm uses the rising flank of the profile to calculate the correction, the presence of the band at ~200 bp strongly interferes with the correction procedure and leads to an aberrant corrected profile.

Figure C. Screenshot of the TeloTool interface with one of the rare samples that could be properly analyzed by TeloTool. The graph represents the analysis of the first lane. The raw data and its fit (in black) closely match the corrected data and its fit (in red), suggesting that correction for unequal probe intensity was negligible. Notice how the multimodal profile (3rd lane from the right) is analyzed without taking into account the multimodality.

- The putative CrTERT gene has 28 introns - can the authors comment on whether this is a high or normal intron number for genes in this organism?

We believe the gene model for *CrTERT* as currently annotated in the genome is wrong, since a gene containing 28 introns is highly unusual in *Chlamydomonas*, although not impossible. The gene model encodes for a 5019-aa long protein, which is also quite unlikely given the size of telomerase catalytic subunits in other plant species (1100-1300 aa). We now comment on this in the result section.

- The homologies found in CrTERT with RNA binding domains and reverse transcriptase domains from other organisms along with the telomere shortening phenotype seen in the disruption strain are indeed suggestive that this gene is CrTERT, but it's not out of the question that this gene is a telomere length regulator rather than the enzyme itself, or that there are multiple telomerase encoding genes. It would be ideal for the authors to perform

TRAP assays on the wild type and disruption strains, as would mutation of the putative RT catalytic residues.

The development of a TRAP assay working for *Chlamydomonas* extract is currently underway but has not been successful and robust yet. This might be due to barely detectable telomerase activity in *Chlamydomonas* (Fulneckova et al. 2013). Thus unfortunately, we cannot reinforce our results with TRAP assay measurements at this stage.

We agree that mutations targeting catalytic residues of *CrTERT* would be ideal and we plan to do it in the future. However, introducing point mutations in *Chlamydomonas* is not straightforward. We have been working on CRISPR/Cas9-based genome editing methods but have yet to make it work robustly.

- The authors do not mention whether mutations can be readily introduced in this organism, so this reviewer cannot be sure whether the latter experiment is possible. If not, could wt and RT-mutant alleles be introduced into the tel-m1/2 mutant cells? Please include in the Introduction a description of what genetic tools are available in *C. reinhardtii*, as well as basics like the number of chromosomes.

Classic and reverse genetics approaches can be used in *Chlamydomonas*. But while some forward genetic tools are available in *Chlamydomonas*, they are nowhere near the efficiency and the versatility of the tools used, for example, in *S. cerevisiae*. For instance, the mutants we describe in the manuscript come from a library of mapped insertional mutants (Li et al. 2016, *The Plant Cell*), a unique and important resource for the community (a second version of the library has just been published: Li et al. 2019, *Nat Genet*). But the insertion of a DNA fragment in a targeted locus in the nuclear genome remains difficult, even with a selection marker. CRISPR/Cas9-mediated genome edition technology is still emerging for this organism. Furthermore, plasmids cannot be stably maintained in the nucleus. Therefore, the experiment suggested by the reviewer has not been performed. The closest we could get from a functional complementation experiment was the backcross of tel-m1/2/4 with a wild-type strain to show the 2:2 co-segregation of the telomere shortening phenotype and the paromomycin marker inserted in the *CrTERT* locus.

We have now added some more information on *Chlamydomonas* as a model organism in the introduction. There are also other information about this organism elsewhere in the manuscript, for example about its physiology and heterotrophy.

- Fig 5B is difficult to evaluate because of the high level of background hybridization - is there a high level of DSBs or are there dying cells, or can a new gel be provided?

We have changed the overall contrast and luminosity of the image, which were initially set to highlight the presence of additional bands and higher apparent molecular weight DNA fragment. We have not evaluated the level of DSBs in these cultures, since methods to study the DNA damage response are limited in *Chlamydomonas*. No obvious cell mortality was observed in these cultures, but we are working on a more detailed characterization of telomerase-independent survivors as a follow-up to this study.

- The definition of 'mt' in Figs 5C and 5D is hard to find, if anywhere.

We apologize for this oversight. “mt” corresponds to the mating type, + or -. When referring to a strain, it is common to add the mating type after the name of the strain (e.g. T222+). For crosses, we decided to explicitly write “mt+” and “mt-” next to the name of the strain. We have now added the definition in the figure legends of figure 5, supplemental figure S4 and S5.

- Assuming that CrTERT is indeed the sole telomerase RT, it appears that survivors of telomerase deletion arise readily, which is interesting. The experiments in figure 5C/D in which cells with and without telomerase are mated and telomere length in the offspring followed with time are useful, but lack crucial information about telomere length in the diploid. If the authors could show that telomere length is restored in the diploid, and erodes over time (more time points than just 21 days and 42 days), an 'ever shorter telomeres' phenotype could be nicely documented; the two time points shown do not outline an EST phenotype. This would shore up the idea that CrTERT is telomerase. Does the CrTERT gene vary at all between isolates (as determined by PCR and sequencing) that have varying telomere length setpoints?

The diploid state of *C. reinhardtii* is obtained by mating two haploid cells of opposite mating types but is naturally a dormant and resistant state called zygospore, with no cell proliferation. Diploid cells then undergo meiosis in the presence of nitrogen and produce haploid cells that can resume vegetative growth. Thus, we could not grow diploid cells and could not even measure their telomere length because of the low number of cells. We could only measure telomere length in the meiotic progeny after dissection of the four haploid cells. The earliest time point that we could harvest with a sufficient number of cells was 21 days after dissection (starting with a single cell on plate). We assumed that in the four haploid cells produced by meiotic division, the average telomere length would be similar since meiosis should have shuffled telomeres independently of the status of *CrTERT*. Thus, the difference in telomere length between the four cultures after 21 days is already indicative of telomere length shortening in the telomerase-negative progeny since the meiosis event. Nevertheless, because of the technical limitation on analyzing the diploid state, we understand the reviewer's comment that the “ever shorter telomeres” phenotype, as first described for yeast, is not sufficiently supported by the evidence. We now qualify this phenotype simply as “telomere shortening”.

As stated above, the current gene model of *CrTERT* is 25 kb long and is most probably wrong. We are thus not yet able to amplify and sequence the entire gene, which would indeed be of great interest. The PCR primers we managed to design to verify the mutants only amplify a short region of ~300 bp. Correcting the gene model is currently one of our goals.

- In Figure S4, please label the diagrams on the left with numbers of bp. Ideally a long-template polymerase would be used to confirm the predicted long products in the 'no amplification' lanes.

The diagrams on the left are used as schemes to illustrate the PCR strategies for the three mutants, irrespective of where the selection cassette is actually inserted. They are not intended to be to scale. We put the sizes of the expected PCR products above each gel. The lanes in Fig. S4A where no amplification could be detected (as expected) were confirmed in Fig. S4B using a primer within the selection marker, thus amplifying a PCR product and demonstrating that the marker was indeed present as expected.

- Check for typos and grammatical errors. The telomeres should be referred to as 'degenerate', not 'degenerated' - those words have different meanings. 'Discreet' should be spelled 'discrete' for this meaning. Some lines, like line 351, should be checked for clarity.

We apologize for the typos and grammatical errors and thank the reviewer for pointing them out. We have proofread the revised version of the manuscript.

- A Discussion that places the results in an evolutionary context more clearly would enhance the interest of the paper, for instance discussing the variability in using telomerase as a length regulator across eukaryotes. Is telomere length variable among different isolates in other species so far studied? Do the authors propose a biological rationale for high variability in *C. reinhardtii*?

We now refer in the discussion to studies reporting natural variations in telomere length in different strains/ecotypes/isolates in other organisms. The relative variability is difficult to compare across species and would require proper normalization regarding genetic divergence within a species, but all these studies point to a complex regulation of telomere length through many genes as well as external cues. We think that there is no strong selective pressure for setting a precise telomere length or distribution and thus telomere length regulation can have different set points in different strains with minor effect on physiology.

We also comment of the discovery of the blunt end structure in *C. reinhardtii*, as it was previously thought to be restricted to angiosperms (Kazda et al. 2012).

Reviewer #2 (Comments to the Authors (Required)):

The manuscript by Eberhard and colleagues sets the stage for a better understanding of telomere biology in the alga *C. reinhardtii*. The authors show that: 1) telomeres comprise mostly non degenerate T4AG3 repeats, which confirms previous work; 2) telomeres are likely not wrapped around canonical nucleosomes; 3) telomeres end with a G-rich overhang; 4) telomere length within the same strain is not affected by culture conditions, while it varies substantially amongst different strains; 5) telomerase maintains telomere length in the alga, and telomerase ablation leads to telomere shortening followed by senescence and possible appearance of ALT-type survivor cells.

This study is interesting, and although it does not present any major conceptual advance in telomere biology, the study of this specific model organism is highly relevant. The data are well presented and experiments are sound. The manuscript is well written. While in principle I support its publication, a few issues should be considered and fixed.

We thank the reviewer for the constructive comments, which helped improving the manuscript. Hopefully, the revised version addresses the issues that the reviewer raised.

1) The PETRA experiments need to be better controlled, as a minimum genomic DNA samples pre-treated with ExoI should be included. Also, while PETRA represent a necessary choice for telomeres with very short overhangs, there is no evidence that this is the case in *C. reinhardtii*. Native TRF in gel hybridization should be performed as a less indirect way to detect overhangs. Also here, ExoI controls should be included, and possibly parallel

hybridizations with G-rich telomeric probes should be performed to test for the existence of C-overhangs, which exists in other species.

We have now added PETRA experiments that include the ExoI-treated control and another negative control that lacks the first primer extension step by the phi29 polymerase. ExoI-treated sample still showed some weak signal, which was to be expected since PETRA is an amplification-based technique, but the decrease of signal upon Exo-I treatment is apparent.

Following the reviewer's suggestion, we now also provide native in-gel hybridization evidence for 3' overhangs, which includes ExoT (equivalent to ExoI) control and both G-strand and C-strand hybridization (Fig 1C and Fig S1D). The in-gel experiment confirms the presence of a 3' overhang and no detectable 5' overhang. The specific experiment shown in Fig 1C had to be cropped since one S24- sample was under loaded and thus hard to compare quantitatively. The uncropped images are shown in Fig S1D for clarity. However, we did observe 3' overhangs in strain S24-, similarly to T222+, in another independent experiment (see below Figure D).

Figure D. In-gel hybridization assay on strains T222+ and S24-. "PDN" correspond to pre-denatured samples (10 min at 95°C just before loading).

Figure 1 and S1 have been modified to include the controls for the PETRA assay and the native in-gel experiment. Appropriate figure legends and text describing the result have been added.

2) The 200 bp intrachromosomal telomeric repeat band seems to be sensitive (albeit less than telomeres) to Bal31 treatment, contrarily to what the authors state.

We agree with the reviewer that with the longest Bal31 treatment, the ~200 bp becomes fainter. This is probably due to some overall degradation of bulk DNA with longer treatment by Bal31 (see below in Figure E the ethidium bromide stained gel corresponding to Figure S3B). However, in contrast to the telomeric signal, the size of the ~200 bp is perfectly maintained in all conditions, thus indicating that the band is indeed internal. We have changed the text to clarify that the migration, specifically, of the band was not altered.

3) In Figure 3B, I am not convinced that the shorter telomere length appearing at day 8 is representing telomere dynamics; it could simply be a partial degradation of that specific DNA sample. How reproducible is this fluctuation in length? Can the authors show also a shorter exposure or less contrasted image in order to better appreciate the integrity of the 200 bp band?

The slight variation at day 8 is not reproducible and Figure 2B was not intended to show this dip in telomere length, but rather that telomere length is relatively stable even after days in saturation. We now show another TRF where this variation is absent.

4) In Figure S3B, something went wrong with lane 2 from the left. Is there much less DNA loaded? Also here it seems that DNA degradation might confound the results. Is there a minuscule asterisk pointing to that lane (I don't see any explanation in the legend)?

Indeed, we inadvertently omitted to explain that the asterisk means that this sample was somehow degraded. The other samples in this gel seem to be intact, as the ~200 bp does not show much intensity variation. See below in Figure E the ethidium bromide stained gel corresponding to his experiment. We would like to point out that each Bal31 experiment contains two negative controls, one that was directly processed as a normal TRF ("NP", No column Purification) and one that was column-purified after being mock-treated with Bal31 ("0") and then processed for TRF analysis. Here, the "NP" sample still provides a good control. Please see Fig. 1F and Fig. S1E, where we can appreciate that both controls are usually quite similar.

We have now added an explanation in the figure legend.

Figure E. Ethidium bromide stained gel corresponding to Figure S3B. The sample in lane 2 indeed contains much less DNA. The DNA was probably degraded during preparation. As observed on the right side of the gel, bulk genomic DNA is often partially degraded with the longest Bal31 treatment.

5) The discussion is extremely long and would profit from shortening, to assure that important messages of the study are not diluted away. Also in the results sessions several details (how Bal31 or TRF work, for example) could be avoided.

We have now modified the discussion based on all the reviewers' comments and have tried to keep it as concise as possible. While we actually shortened the discussion of the initially submitted version, we also added some new information and discussion, notably about the blunt end structure, thus ending up with a barely shorter discussion.

We have also removed some details regarding TRF and Bal31 digestion in the results section, as suggested.

Reviewer #3 (Comments to the Authors (Required)):

This very careful researched study investigated several key components of telomere biology in *Chlamydomonas reinhardtii*, a photosynthetic unicellular green alga. Telomere sequence

and structure are analyzed in detail, which confirmed the TTTTAGGG repeat sequence identified previously by Berman and colleagues, and also showed (as in the case for most species) that chromosome termini end with a G-rich overhang. Data is also provided that shows that there is significant variation in steady-state telomere length among different *C. reinhardtii* strains; although the extent of variation is striking, the result itself is not novel, given the dramatic length differences among (for example) closely related inter-fertile species of mice.

This structural analysis was combined with genetic analysis of three pre-existing CrTERT insertional mutations, thereby confirming the prior identification of the genomic locus of this gene. The resulting telomerase-defective strains exhibited the predicted progressive telomere shortening, as well as a possible survivor phenotype, although the latter was not investigated. Interestingly, crosses between telomerase-null and telomerase-proficient strain, following by meiosis to generate haploid derivatives, resulted in a telomere-length segregation phenotype, indicating that telomerase levels in the heterozygous diploid were not sufficient to re-set telomere length back to wild type levels.

Although the experiments were carefully conducted, the conclusions are largely confirmatory, with no new insights about telomere biology, either in general or in *C. reinhardtii*.

We thank the reviewer for his/her feedback. We believe that a detailed molecular characterization of telomeres in *C. reinhardtii* is important to diversify the spectrum of photosynthetic model organisms used for telomere studies, *A. thaliana* being essentially the only well characterized model. We also think that studying the processes at telomeres will help understand other aspects of nuclear biology, such as the DNA damage response or genome instability, which is of great interest to the *Chlamydomonas* community.

We have now found that a subset of *Chlamydomonas* telomeres can be blunt-ended, suggesting that blunt-end structures might have an ancient origin, as they are also present in *A. thaliana*. This discovery also opens the question of the factors implicated in the protection of such blunt ends in comparison to the more widespread 3' overhang structure, with the Ku complex being a prime candidate.

May 21, 2019

RE: Life Science Alliance Manuscript #LSA-2019-00315-TR

Dr. Zhou Xu
CNRS - Sorbonne Université
Institut de Biologie Paris Seine, Laboratoire de Biologie Computationnelle et Quantitative UMR7238
4 place Jussieu
75252 Paris
France

Dear Dr. Xu,

Thank you for submitting your revised manuscript entitled "Molecular characterization of *Chlamydomonas reinhardtii* telomeres and telomerase mutants". As you will see, the reviewers appreciate the changes introduced and we would thus be happy to publish your paper in Life Science Alliance, pending final revisions:

- please incorporate the suppl methods into the main manuscript file - you can also add the STable and Sfigure legends to the main manuscript file
- please add a callout to Table S1 (primers used)
- please also check one more time all your manuscript files and experimental data (see also comment of reviewer #2) and upload final files

A. FINAL FILES:

-- Summary blurb (enter in submission system): A short text summarizing in a single sentence the study (max. 200 characters including spaces). This text is used in conjunction with the titles of

papers, hence should be informative and complementary to the title. It should describe the context and significance of the findings for a general readership; it should be written in the present tense and refer to the work in the third person. Author names should not be mentioned.

B. MANUSCRIPT ORGANIZATION AND FORMATTING:

Sincerely,

Andrea Leibfried, PhD
Executive Editor
Life Science Alliance
Meyerohofstr. 1
69117 Heidelberg, Germany
t +49 6221 8891 502
e a.leibfried@life-science-alliance.org
www.life-science-alliance.org

Reviewer #1 (Comments to the Authors (Required)):

The additional experiments and explanations provided have addressed most of the concerns raised.

Reviewer #2 (Comments to the Authors (Required)):

The authors have addressed quite well all my criticisms. Honestly, the fact that repetitions of experiments now reveal something different compared to the original version is a worrisome, specifically because the authors state that the original results were not correct due to the quality of the material used (genomic DNA preps). I would strongly encourage the authors to make sure that everything else is correct. If that is the case, I support publication of this manuscript.

May 27, 2019

RE: Life Science Alliance Manuscript #LSA-2019-00315-TRR

Dr. Zhou Xu
CNRS - Sorbonne Université
Institut de Biologie Paris Seine, Laboratoire de Biologie Computationnelle et Quantitative -
UMR7238
4 place Jussieu
75252 Paris
France

Dear Dr. Xu,

Thank you for submitting your Research Article entitled "Molecular characterization of *Chlamydomonas reinhardtii* telomeres and telomerase mutants". It is a pleasure to let you know that your manuscript is now accepted for publication in Life Science Alliance. Congratulations on this interesting work.

*****IMPORTANT:** If you will be unreachable at any time, please provide us with the email address of an alternate author. Failure to respond to routine queries may lead to unavoidable delays in publication.*******

DISTRIBUTION OF MATERIALS:

Again, congratulations on a very nice paper. I hope you found the review process to be constructive and are pleased with how the manuscript was handled editorially. We look forward to future exciting

submissions from your lab.

Sincerely,
